# "Whoever Harms a Dhimmī I Shall Be His Foe on the Day of Judgment": An Investigation into an Authentic Prophetic Tradition and Its Origins from the Covenants

Ahmed El-Wakil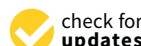

College of Islamic Studies, Hamad Bin Khalifa University, P.O. Box 34110 Doha, Qatar;
ahmed.a.elwakil@gmail.com

**Abstract:** The ḥadīth, "whoever harms a dhimmī I shall be his foe on the Day of Judgment', can be found as an end clause to covenants which the Prophet Muḥammad issued to Christian, Jewish, and Magian communities. As it is highly unlikely for different non-Muslim communities to have forged this Prophetic statement at the end of their respective documents, this paper argues that this utterance is authentic and can be confidently traced back to the Prophet. This paper examines the occurrence of this statement as a ḥadīth in the Islamic literature and notes how it was dismissed by scholars of tradition who only accepted one of its variants. The paper then compares the rights granted to non-Muslims in the covenants to those conveyed in a number of ḥadīths and notes the discrepancies between early Islam's official documents and the legal injunctions found in Muslim tradition. It argues that the ḥadīths on the rights of non-Muslims oftentimes reflect legal maxims of scholars living in the 'Abbasīd era and that these were back-projected to the Prophet and his Companions using fictitious isnāds. Finally, this paper concludes by recommending the incorporation of the Prophet's official decrees, which includes the covenants, within the fabric of Islamic law.

**Keywords:** Prophet; Muhammad; Islam; covenant; authenticity; dhimmi; hadith

## 1. Introduction

Muhammad Hamidullah's referential work *Majmūʿat al-Wathāʾiq al-Siyāsiyya li-l-ʿahd al-Nabawī wa-l-Khilāfat al-Rāshida* brought together all the letters and political treaties of the Prophet and the Rightly Guided Caliphs in one volume spanning just over 750 pages. The sheer magnanimity of traditions reporting how official decrees were issued during the rise of Islam cannot all be based on fabrication. As a matter of fact, the authenticity of the *Constitution of Madīna* reveals how the Prophet began writing political decrees early on and that this practice was continued by his Companions after his death. In an attempt to shed light on the nature of these official decrees and the motivation behind them, this paper adopts an inter-textual examination of what Muslim and non-Muslim sources report about them.

As Jewish, Samaritan, Christian, and Magi traditions all unanimously agree that their communities received a covenant of protection from the Prophet guaranteeing their safety and religious liberty, their claims cannot be disregarded without a careful investigation. The question that therefore arises is not whether the Prophet issued official decrees, for this should be regarded as historical fact, but rather where are these official decrees now, and more importantly, in the absence of the originals, what was their content?

Past scholarship on the covenants looked at these in silos, leading scholars to either reject their authenticity or at best express skepticism over their provenance. Ever since the publication of John

Andrew Morrow's book *The Covenants of the Prophet Muḥammad with the Christians of the World*, there has been renewed interest in these documents. The obvious "mistakes" which led to the rejection of the covenants, such as their early dating, the names of witnesses, and the scribe's name, have all proved to be consistent anachronisms in texts inherited by non-Muslim communities for which there is no evidence of cross-communal borrowing or influence. Analysis of these textual parallelisms have, thus, led to a counterintuitive conclusion, namely, that the covenants are authentic and, on the whole, textually accurate but that they were deliberately suppressed or interpolated in Muslim sources.[1] Though one is not here arguing that all letters and official decrees of the Prophet and the early Caliphs that exist in Muslim tradition are edited versions of the originals, one nevertheless cannot negate the possibility of a policy having developed around the 2nd/8th century to undermine and reverse the rights that were initially granted to non-Muslim communities.

## 2. A Mutawātir Dictum in the Covenants

The term dhimmī in Arabic means "a protected person" or a "person who has been granted a pact of protection" and usually refers to a non-Muslim citizen of a Muslim nation. An analysis of the covenants reveals that the tradition "whoever harms a dhimmī I shall be his foe on the Day of Judgment" can be found at the end of a number of covenants in the possession of different non-Muslim communities. Considering that a mutawātir report can briefly be defined as one which has been transmitted by a large enough number of people who could not have possibly colluded or agreed to fabricate it,[2] any fair-minded observer would certainly consider it very peculiar for different and geographically dispersed non-Muslim communities to have come together and conspired to forge documents which would have included this clause at the end of them on the authority of the Prophet. When we complement this observation with internal and external evidence adjudicating the authenticity of the covenants, it renders the allegation of forgery highly unlikely.

The first Prophetic covenant to incorporate this clause was from a manuscript recorded by Michel Gabriel in his book *Tārīkh al-Kanīsa al-Anṭākiyya al-Suryāniyya al-Mārūniyya,* first published in 1899 CE.[3] Another copy of the exact same covenant was discovered by a French officer who participated in Napoleon Bonaparte's expedition to Egypt and which was published in Jeanne Aubert's *Le Serment du Prophète* in 1938 CE.[4] The Gabriel recension reads that "it was written to all Christians and to all the places where Christians reside."[5] The text of the Aubert recension is somewhat faint, but appears to read that it was addressed "to all Jacobite Christians and all places [which they reside] (li-kāffat al-naṣārā al-yaʿāqib li-sāʾir al-amākin)."[6] The covenant was scribed by ʿAlī b. Abī Ṭālib on 11 Muḥarram 2 AH[7] and its last clause reads:

> "Whoever afterwards commits an injustice towards a protected person by breaking or rejecting the covenant, I shall be his foe on the Day of Judgment from among all the Believers and the Muslims."

> "wa man ẓalama baʿd dhalika dhimmīyyan wa naqaḍa al-ʿahd wa rafaḍahu kuntu khaṣmahu yawm al-qiyāma min jamīʿ al-mūʾminīn wa al-muslimīn kāffatan."
>
> [8]

---

[1]　For recent scholarship on the covenants see (Morrow 2013, 2017). Also see (El-Wakil 2016, 2017).

[2]　For a contemporary definition rooted in classical Muslim orthodoxy, see (Keller and al-Misri 1997), w48.2. For the definition given by Abū al-Qāsim al-Balkhī, see (Hansu 2009, pp. 391–92).

[3]　(Gabriel 1900).

[4]　(Aubert 1938).

[5]　(Gabriel 1900), *Tārīkh al-Kanīsa*, p. 588.

[6]　(Aubert 1938). For a copy of a covenant addressed to the Christian Jacobites, see Akyüz 2002.

[7]　(Gabriel 1900), *Tārīkh al-Kanīsa*, p. 594.

[8]　Gabriel, *ibid.*, p. 593; (Aubert 1938), *Le Serment du Prophète*, image between pp. 40–41.

An identical clause can be found at the end of the copy of the Prophetic Covenant stored at the Hill Museum and Manuscript Library.[9]   Another recension of the same covenant was documented by John Andrew Morrow in his 2013 book, *The Covenants of the Prophet Muhammad with the Christians of the World*, the only difference being that the word "al-mū'minīn" is missing and it reads instead "I shall be his foe on the Day of Judgment from among all the Muslims (kuntu khaṣmahu yawm al-qiyāma min jamī' al-muslimīn kāffatan)."[10] Though the Hill/Morrow Covenant does not address any particular Christian sect, it appears that it was either written to al-Sayyid Ghassānī, the governor of Najran, or to the Copts of Egypt.[11]

Georg Graf discovered a different Prophetic covenant of which a similar recension was documented by Father Gabriel Akyüz[12] of the Syriac Orthodox Church of Antioch and which was addressed "to all Christian sects and to the Copts of Egypt and all the provinces there."[13] The Graff/Akyüz recension reads at the end:

> "Whoever falls short of upholding the conditions [of this covenant], goes contrary to them, or does not abide by my instructions, and so, as a consequence mistreats a protected person, I shall be his foe in front of Allah on the Day of Judgment and so will all the believing men and women in their entirety."

> "wa man naqaḍa wa 'amila bil-khilāf wa lā sami'a kalāmī wa ẓalama dhimmīyyan anā akūn khaṣmahu quddām Allāh yawm al-qiyāma wa jamī' al-mū'minīn wa al-mū'mināt kāffatan ajma'īn." [14]

Though the name of the Prophet's scribe missing in the Graf/Akyüz recension, Louis Cheikho reported in the recension, which he had come across, that the Prophet had instructed Mu'āwiya to write it.[15] The recension reported by Cheikho was written to "the Copts and the Syriac Jacobites of Egypt and its provinces."[16] The slight differences in the last clause of the version consulted by Cheikho to the Graf/Akyüz recension are highlighted in bold below:

> "wa man naqaḍa wa 'amila **bi-khilāf al-shurūṭ** [instead of **bil-khilāf**] wa lā sami'a kalāmī wa ẓalama dhimmīyyan akūn khaṣmahu yawm al-qiyāma [instead of **anā** akūn khaṣmahu **quddām Allāh** yawm al-qiyāma]." [17]

The obvious question that comes to mind is whether the recensions of (1) Gabriel/Aubert, (2) Graff/Akyüz/Cheikho, and (3) Hill/Morrow are in fact three recensions of the same covenant. A cross-comparison of the witnesses' names reveals up to 16 identical names for all of these three covenants. The Gabriel/Aubert Covenant has 18 identical names to the Graff/Akyüz recension and 22 identical names to the Hill/Morrow Covenant, while the Graf/Akyüz recension has 19 identical names to the Hill/Morrow Covenant. All three covenants have a total of 30 witnesses' names (see Appendix A).

It is certainly difficult at this point in time to come up with any definitive conclusions, especially when we take into consideration the similarities between the Gabriel/Aubert and Hill/Morrow covenants;

---

[9]   (Ar. 202 2008). The Covenant is on pages 155b–162a. The Prophetic warning is on p. 161b.

[10]   (Morrow 2013, p. 252).

[11]   (Ar. 202 2008), p. 155b reads "katabahu lil-sayyid" which may be an allusion to al-Sayyid Ghassānī who received a number of covenants from the Prophet and which he distributed to other Christian denominations. As for the Morrow recension which was discovered in Egypt, it reads "katabahu al-Asad" being a reference to 'Alī b. Abī Ṭālib (see Morrow, *ibid.*, *Covenants*, p. 255). For more background information see (El-Wakil 2016).

[12]   See (Akyüz 2002), *Osmanlı Devletinde Süryani Kilisesi*, pp. 158–61.

[13]   (Graf 1914), "Ein Schutzbrief Muḥammeds für die Christen," p. 562. (Akyüz 2002), *Osmanlı Devletinde Süryani Kilisesi*, p. 159.

[14]   (Graf 1914), "Ein Schutzbrief Muḥammeds für die Christen," p. 565; (Akyüz 2002), *Osmanlı Devletinde Süryani Kilisesi*, p. 160.

[15]   (Cheikho 1909), *Uhūd Nabī al-Islām*, p. 617.

[16]   Cheikho *ibid*.

[17]   Cheikho, *ibid*.

however, one nevertheless suspects that all three covenants are independent of one another. The date in the Gabriel recension may also give us a clue as to why more than one covenant may have been issued to the people of Egypt at around the same time. The Gabriel/Aubert Covenant was issued on 11 Muḥarram 2 AH/15 July AD 623[18] eight days after the *Covenant with the Monks of Mount Sinai* which is dated 3 Muḥarram 2 AH/7 July AD 623.[19] The Persians invaded Egypt in 618 CE and it may well be that various Christian communities in Egypt who knew Muḥammad from his days as a merchant, and who prophesized his future as a great king, could have arranged to send delegations to meet with him around that time. Another possibility is that once he had consolidated his power in Madīna, the Prophet had sent emissaries to deliver his covenants to these communities.[20]

The *Covenant with the Jews of Khaybar and Maqnā*, which was discovered by Hartwig Hirschfeld in the Cairo Genizah in 1903 CE, also has a similar expression to the three Christian covenants so far examined. It states:

> "Whoever reads this writ of mine or to whoever it is read out to and he alters or changes anything of what is in it, upon him shall be the curse of Allah and the curse of those who curse from among the angels and all of mankind. Such a person is free from my protection and intercession on the Day of Judgment and I am his foe. Whoever is my foe is the foe of Allah, and whoever Allah has declared as foe goes to hell."

> "wa man qarā' kitābī hadha aw quri'a' 'alayhi wa ghayara aw khālafa shay'ān mimmā bihi fa-'alayhi la'natu Allāhi wa la'nat al-lā'inīn min al-malā'ika wa al-nās ajma'īn wa huwa bari'un min dhimmatī wa shafā'atī yawm al-qiyāma wa anā khaṣmuhu wa man khāṣamanī fa-qad khāṣama Allāh wa man khāṣama Allāh fa-huwa fī al-nār." [21]

The statement is also repeated in the midst of *the Covenant with the Children of Israel*:

> "Whoever commits an injustice towards a protected person, even if it by an atom's weight, Allah shall not bless that which is in the possession of his right hand nor his lot and fortune, and I shall be his foe [literally, an advocate against him] on the Day of Judgment. Whoever harms them, harms me, and he who wrongs them wrongs me, and I shall be his foe on the Day of Reckoning and punishment, the day in which he will enter his grave alone."

> "wa man ẓalama dhimmī mithqāl dhara fa-lā bārak Allāhu lahu fī mā malakat yamīnuhu wa fī ṣaybihi wa naṣībihi wa kuntu anā ḥajījuhu yawm al-qiyāma. wa man ādhāhum ādhānī wa man ẓalamahum ẓalamanī wa anā akūn khaṣmahu yawm al-ḥisāb wa al-'iqāb yawm yadkhul qabrahu waḥdahu. [22]

We also find at the end of the text the Prophet's warning to the Muslims:

---

[18]　(Fourmilab n.d.).

[19]　(Hamidullah 1987). The Julian date was derived from the (Fourmilab n.d.).

[20]　The early dating of the covenants is certainly problematic, but these dates should nevertheless be accepted as genuine despite the lack of historical data to justify them. The Prophet had good relations with the Christians of Najran from the time he was in Makkah, and it appears that the latter played a key role in conveying his covenants to the main mother churches which were represented in South Arabia i.e., Miaphysite, Chalcedonian, and Nestorian. We also have two Christian texts prophesizing Muḥammad's future by monks who either visited or resided in Mount Sinai. The first is the apocalypse of Sergius Baḥīra who was shown the future of the Arab empire by an angel who visited him on Mount Sinai. See (Roggema 2009). The second is a text reported by Aḥmad Zakī Basha and recorded by Hamidullah in which a monk residing on Mount Sinai and knowledgeable in astrology accurately prophesised the future of the young Muḥammad. See (Hamidullah 1987), *Majmūʿat al-Wathāʾiq al-Siyāsiyya*, p. 565. Though deemed credulous in our day and age, prophecies, visions, and miracles were taken very seriously in late antiquity's cultural milieu.

[21]　(El-Wakil 2017), "Searching for the Covenants," sct. 41, pp. 112–13. Also see (Hirschfeld 1903). Hirschfeld's translation was edited by me.

[22]　(Ahroni 1998, pp. 78–79). Ahroni's translation was edited by me. For a slight variant where the verb "khaṣama" is used, see (Rivlin 1935, p. 152). The expression therein reads "kuntu anā ḥajījahu wa khaṣmahu yawm al-qiyāma." Also see

> "Whoever shows enmity towards them, then he has shown enmity towards me and towards Allah, may He be exalted. The Lord—exalted be He—and <u>I shall be his foe on the Day of Judgment</u>."

> "man 'ādāhum 'adānī wa 'adā Allāhu subḥānahu wa anā wa al-rab subḥānahu <u>akūn ḥajījahu yawm al-qiyāma</u>." [23]

The recension of the *Covenant with the Magi* brought to light by Sorabjee Jamshetji Jejeebhoy states:

> "Whoever does them harm does me harm and <u>I shall be his foe on the Day of Judgment</u>, his recompense will be the fire of hell and he will be free of my protection."

> "wa man ādhāhum ādhānī wa anā <u>khaṣmuhu yawm al-qiyāma</u> wa jazā'uhu nāru jahanam wa bari'at minhu dhimmatī." [24]

It is noteworthy that the expression "wa man ādhāhum ādhānī" in the Prophet's *Covenant with Magi* can be found almost identically in Rivlin's recension of the *Covenant with the Children of Israel* when it reads "wa man ādhāhum fa-qad ādhānī."[25] It is significant that a text very similar to that of Jejeebhoy discovered "on an old scroll"[26] by the Shī'a scholar Mirzā Ḥusayn al-Nūrī Ṭabarsī (died 2 CE/1320 AH) and published in full in his book *Kalima Ṭayyiba*[27]—and later reproduced by Mūsā b. 'Abdullāh al-Zanjānī (died 1979 CE/1399 AH) in his work *Madīnat al-Balāgha*[28]—suggests that Jejeebhoy, Ṭabarsī, and al-Zanjānī all inherited their versions from an original Parsi source.

## 3. Early Testimonies Attesting the Authenticity of the Covenants

The earliest allusion to the covenants is an inscription discovered in Jerusalem and recently studied by Moshe Sharon which he describes as "probably one of the oldest inscriptions hitherto uncovered."[29] It reads:

1. In the name of Allah, the Most Gracious, the Most Merciful
2. …
3. …
4. The protection of Allah and the security of His messenger (*dhimmat Allāh wa ḍamān rasūlihi*)
5. …
6. It was witnessed by 'Abd al-Raḥmān b. 'Awf
7. al-Zuhrī and Abū 'Ubayda b. al-Jarrāḥ.
8. Its scribe is Mu'āwiya …
9. The year thirty-two.[30]

The parallels between the covenants and the "Jerusalem 32" inscription are astonishing. The statement that the people of Jerusalem have been granted "the protection of Allah and the security of

---

(Goitein 1993, p. 509). The verbal noun "khaṣmahu" is omitted and the phrase reads instead "wa lā bārak Allāhu lahu li-man ẓalama banī isrā'īl mithqāl dhara wa anā ḥajījuhu yawm al-qiyāma". Also see (Nini 1983, p. 196). Nini reads the last clause as "fa-innanī khaṣīmahu wā ḥajījahu yawm al-dīn, yawm al-Ākhir." One criticism of the Covenant with the Children of Israel is that the Imām should be of the progeny Fāṭima, giving us the impression that it was written during Zaydī rule. However, this may have been an explanatory note to the original covenant which was added to it by the Jews of Yemen. It certainly does not affect the overall authenticity of the text.

23 Ahroni, *ibid.*, pp. 88–89. Ahroni's translation was edited by me.
24 (Jejeebhoy 1851). Also see (El-Wakil 2017) "Searching for the Covenants," sct. 12, p. 127.
25 (Rivlin 1935), "Ṣava'at Muḥammad le-'Alī ben Abī Ṭālib," p. 152.
26 (Ṭabarsī 1988).
27 (Ṭabarsī 1381 AH).
28 (al-Zanjānī 1343 AH).
29 (Sharon 2018, p. 109).
30 Sharon, *ibid.*, p. 101.

His messenger", the scribal conventions reflected in the presence of witnesses, Muʿāwiya as scribe, and a year date, all conform to the manner in which the Prophetic covenants were written. Taking into consideration how ancient civilizations carved inscriptions in the nature of proclamations to the general public so leaders could set forth the norms which they wanted the people to know and obey, the inscription clearly establishes the terms of the relationship between Islam and the protected people in a lasting and public form. It thereby indicates how the Rightly Guided Caliphs took the Prophet's covenants as a precedent for making such commitment to their subjects in the face of history.

The most palpable written reference to the covenants can be found in John Bar Penkaye's *Universal History*, which he wrote in 67 AH/687 CE when he described how these were implemented as part of state policy during Mu'āwiya's Caliphate. Anyone who has carefully analyzed the texts of the covenants immediately realizes that his descriptions of the early Caliphate fully concord with their contents:

> "A man among them named Mu'āwiya took the reins of government of the two empires: Persian and Roman. Justice flourished under his reign, and a great peace was established in the countries that were under his government, and allowed everyone to live as they wished. They [i.e., the Muslims] had received, as I said, from the man who was their guide [i.e., Prophet Muḥammad], an order [i.e., a covenant] in favour of the Christians and the monks ... Of each person, they required only tribute [i.e., the jizya] allowing him to remain in whatever faith he wished ... While Mu'āwiya reigned there was such a great peace in the world as was never heard of, according to our fathers and our fathers' fathers ... [31] ... There was no difference between pagan and Christian, the believer was not distinct from the Jew, and did not differ from the deceiver." [32]

The covenants are also indirectly referred to by another near contemporary of the Prophet, the Catholicos Isho'yahb III (died 659 CE) who expressed how Muslims are "no enemy to Christianity, but they are even praisers of our faith, honorers of our Lord's priests and holy ones, and supporters of churches and monasteries."[33] 'Abdullāh b. Isḥāq b. Ismā'īl al-Hāshimī later referred to them in his epistle which he wrote to 'Abd al-Masīḥ b. Isḥāq al-Kindī during the reign of the Caliph al-Ma'mūn around 830 CE.[34] There are a number of clues from his epistle to indicate that he was indeed referring to no other documents than the covenants. To begin with, al-Hāshimī begins his epistle by conferring the greetings of peace and mercy on al-Kindī based on the practice of the Prophet.[35] He then states that the Prophet "gave them [i.e., the Christians] covenants and pledges (wa a'ṭāhum al-'uhūd wa al-mawāthiq)"[36], an expression which occurs word for word in the covenants.[37] The writer then explains how the Prophet "gave them protection (ja'ala lahum min al-dhimma) as he would to himself, and made his Companions give them protection in the same manner as they would do so to themselves. He wrote for them treaties and decreed this upon them, and he certified this to them when their delegations met him."[38] Al-Hāshimī's comments are clear evidence that this author had indeed read the original documents.

---

[31] (Mingana 1908). I have relied on the English translation of Roger Pearse. See (Penkaye 2010).

[32] (Mingana 1908), *ibid.*, p. 179. English translation by Roger Pearse.

[33] (Penn 2015).

[34] See (Muir 1887).

[35] (Tartar 1997). I was able to locate a copy of the letter of 'Abdullāh al-Hāshimī and the response of al-Kindī appended to it in a leather binding completed on Sunday 22 Ṣafar 1093/1 March 1682 in the Biblioteca Nazionale Marciana in Venice. See "Dialogus de rebus Fidei Christianum Mohametanum," Cod. XIV/MSS. Orientali No. 14, ff. 113, Collocazione 109. This manuscript appears to have been unknown to Tartar.

[36] (Tartar 1997), *Ḥiwār Islāmī*-Masīḥī, p. 11.

[37] To read the reconstructed Master Template, see (El-Wakil and Nasrallah 2017). The exception to this is the Prophet's Covenant with the Monks of Mount Sinai. This expression is in the singular form in the Morrow Covenant. Also see Gabriel, *Tārīkh al-Kanīsa*, p. 593.

[38] (Tartar 1997), *Ḥiwār Islāmī-Masīḥī*, p. 35. For more details see John Andrew (Morrow 2017), "The Provenance of the Prophet's Covenants," pp. 185–88.

We find the earliest recension of the Prophet's *Covenant with the Magi* in Abū al-Shaykh al-Iṣfahānī's (died 979 CE/369 AH) *Ṭabaqāt al-Muḥaddithīn bi-Iṣfahān*. [39] A copy of this document can also be found in Abū Nuʿaym's (died 1038 CE/430 AH) [40] *Dhikr Akhbār Iṣfahān* and in al-Sayyid ʿAlī Khān al-Shīrāzī's (died 1708 CE/1120 AH) *al-Darajāt al-Rafīʿa fī Ṭabaqāt al-Shīʿa*[41] of which a summary of its contents was made much earlier by Ibn Shahrashūb (died 1192 CE/588 AH) in *al-Manāqib*.[42] It should be noted here that all of these recensions include the Prophet's warning that on the Day of Judgment, he will be the foe of whoever harms them (Ibn Shahrashūb's textual summary merely alludes to it). Both Abū al-Shaykh and Abū Nuʿaym recall that the covenant was given to the Magi and that it was kept in the possession of Ghassān b. Zādhān, one of Salmān al-Fārisī's descendants. Though this document was first recorded in the 4th/10th century, we know from Ibn Shahrashūb that it continued to be in circulation in the 6th/12th century when he comments that "the writ is up to this day in their hands [i.e., with the Magi] and the people abide by the decree of the Prophet, peace and blessings be upon him and his family."[43]

Historians of the middle-ages were clearly aware of the covenants when they alluded to the Prophet's eternal promise to Christians. Samuel of Ani (died 1185 CE) explained how, with "an eternal oath he [Muḥammad] sealed a deed for the land of Armenia, [that] they could freely observe Christianity,"[44] a testimony that was later shared by Bar Hebraeus (died 1286 CE) when he explicitly stated how the *Covenant with the Christians of Najran* was an eternal pact to all Christian denominations (decretum ad chrisrianos pertinens).[45] *The Legend of Sergius Baḥīra* also has a brief reference to the covenants in both the Long and the Short Arabic Recensions[46], though it is the Long Arabic Recension which is of particular importance to us here when it has Baḥīra observe:

> "He [Muḥammad] said to me: 'It is my duty to order my people not to take the jizya or kharāj from monks, to respect them and to fulfill their needs and to care for their circumstances. And I will demand from them, with regard to all the Christians, that they do not to act unjustly towards them, and that their ceremonies will not be changed, and that their churches will be built, and that their heads will be raised, and that they will be advanced and treated justly. And whoever is unjust to one of them I shall be his foe on the Day of the Judgment (wa man ẓalama aḥadan minhum kuntu khaṣmahu yawm al-qiyāma).'" [47]

The earliest we can date the Long Arabic Recension is 1306 CE[48], though it is possible it was written at an earlier time for it appears that al-Masʿūdī referred to *The Legend* in the 940s CE.[49] As Barbara Roggema explains, it is an "archetypical counterhistory" which "builds its case primarily on Islamic tales, doctrines and Scripture. It is through its agreement with some key elements of Muslim sources that it tries to convince its audience of its interpretation."[50] Despite being a polemical work, there is no reason why *The Legend* could not have used as inspiration a genuine Prophetic covenant as a reliable source amidst its mythical narrative to explain away the successful Muslim conquests.

The first text to reproduce a Prophetic covenant in its entirety is the 10th century *Chronicle of Seert* which was compiled before "the reign of Ishoʿyahb IV in 1023 [CE]."[51] The *Covenant with the*

---

[39] (al-Iṣfahānī 1992).
[40] (Abū Nuʿaym 1990).
[41] (Sayyid ʿAlī Khān al-Shīrāzī 1397 AH).
[42] (Ibn Shahrashūb 1956).
[43] (Ibn Shahrashūb 1956), *al-Manāqib*, vol. 1, p. 111.
[44] (Thomson 1994, p. 843).
[45] (Barhebraei 1877).
[46] (Roggema 2009). Also see the Long Arabic Recension for a similar reference to the covenants: pp. 526–27.
[47] Roggema, *ibid*., pp. 456–57. Roggema's translation has been edited by author.
[48] Roggema, *ibid*., p. 240.
[49] (Szilágyi 2008, p. 202).
[50] (Roggema 2009) *The Legend of Sergius Baḥīra*, p. 34.
[51] (Wood 2013).

*Christians of Najran* which it records was apparently copied out in conformity to the original document by Ḥabīb the Monk who was a former keeper of manuscripts.[52] Similarly, MS 696—being a copy of the Prophet's *Covenant with the Monks of Mount Sinai* and which is dated to 2 Rajab 968 AH/19 March 1561 CE—states at the end of it that "This copy has been transcribed from the replica that was copied out in conformity to the original covenant and which was handwritten by the Commander of the Believers ʿAlī b. Abī Ṭālib (nuqilat hadhihi al-nuskha min al-nuskha al-latī nuqilat min al-nuskha al-manqūla min al-nuskha al-kāyina bi-khaṭ amīr al-mūʾminīn ʿAlī b. Abī Ṭālib)."[53] Other copies of the *Covenant with the Monks of Mount Sinai*—namely, MS 695 and five scrolls which are part of the Arabic collection at St. Catherine's Monastery, which has been made publicly available by the Library of Congress—also claim that they were copied in conformity to the replica.[54] The *Covenant with the Monks of Mount Sinai* came to officially be recognized by the Ottomans, and evidence of this can be found in how it was reproduced in *Majmūʿat Mansha'at al-Salāṭīn* which was compiled by the Head of the Ottoman Chancery, Ferīdūn Beg (dated 1583 CE/991 AH), in 1575 CE.[55]

Covenants issued to other Christian denominations in the Prophet's time are so similar to the copy given to the monks of Mount Sinai that one can only conclude that the differences in the dates, names of witnesses, and the scribes' names are all derivations that emanated from a Master Template which was in the possession of both ʿAlī and Muʿāwiya.[56] The omission of the Prophet's statement about the gravity of harming a dhimmī in his *Covenant with the Monks of Mount Sinai* is simply reflective of minor stylistic divergences which the scribes could have been at liberty to either use or omit, leading to slight differentiations in all of the Christian covenants.

The descriptions of these covenants are also of note. The original *Covenant with the Magi* is said to have been written "on white leather skin stamped with the Prophet's seal and the seals of Abū Bakr and ʿAlī, may Allah be pleased with them both."[57] Ḥabīb the Monk's description of the original *Covenant with the Christians of Najran* tells us that "it was written on oxhide that had become yellowish, stamped with his [i.e., the Prophet's] seal, peace be upon him."[58] The *Covenant with the Monks of Mount Sinai* is also said to have been "stamped with the seal of the Prophet and it was written on an old leather parchment."[59] The *Covenant with the Children of Israel* states "my seal and the date, serve as a testimony to them and to my community until the Day of Resurrection, and as long as I and my community endure."[60] These descriptions indicate that the covenants were official decrees stamped with the Prophet's seal and that they were well-preserved relics which came to be recorded and documented at a later point in time.

If these attestations are not convincing enough, critics would still need to answer the following questions: How have two covenants come to be accurately dated to the day of the week?[61] Why is it

---

[52] (El-Wakil 2016), "The Prophet's Treaty with the Christians of Najran," p. 334.

[53] (Library of Congress MS 696).

[54] The scrolls and MS 695 carry that statement almost identically with small occasional differences. See: Scroll 1, online: https://www.loc.gov/resource/amedmonastery.00279389013-ms/?sp=4; Scroll 2, online: https://www.loc.gov/resource/amedmonastery.00279389013-ms/?sp=7; Scroll 3, online: https://www.loc.gov/resource/amedmonastery.00279389013-ms/?sp=11; Scroll 4, online: https://www.loc.gov/resource/amedmonastery.00279389013-ms/?sp=14; Scroll 5, online: https://www.loc.gov/resource/amedmonastery.00279389013-ms/?sp=18; MS 695, https://www.loc.gov/resource/amedmonastery.00279391500-ms/?sp=13 (accessed on 15 November 2017).

[55] (Beg, Ferīdūn 1848–1958).

[56] (El-Wakil and Nasrallah 2017), "The Prophet Muḥammad's Covenant with the Armenian Christians," pp. 472–76, 487–505.

[57] (al-Iṣfahānī 1992), *Ṭabaqāt al-Muḥadithīn bi-Iṣfahān*, vol. 1, p. 231; (Abū Nuʿaym 1990), *Dhikr Akhbār Iṣfahān*, vol. 1, p. 79.

[58] (Scher [1918] 1983, p. 281).

[59] (Hamidullah 1987), *Majmūʿat al-Wathāʾiq al-Siyāsiyya*, p. 563. Also see MSS 695 and 696 as well as the five scrolls in St. Catherine's Monastery.

[60] (Ahroni 1998), "Some Yemenite Jewish Attitudes towards Muḥammad's Prophethood," pp. 50, 88–89. Also see (Rivlin 1935), "Ṣavaʾat Muḥammad le-ʾAlī ben Abī Ṭālib," p. 155; (Goitein 1993), "Kitāb Dhimmat an-Nabī," p. 509; (Nini 1983), "Ketav ḥasūt la-Yehūdīm ha-meyūḥas lannavī Muḥammad," p. 196.

[61] (El-Wakil 2016), "The Prophet's Treaty with the Christians of Najran," pp. 331–32. The Covenant with the Christians of the World was written on Monday 29 Rabīʿ al-Thānī 4 AH and the Covenant with the Jews of Khaybar and Maqnā would have been on Friday 3 Ramaḍān 9 AH.

that all of them seem to abide by the same scribal conventions, possessing the name of a scribe, a date, and names of witnesses?[62] How is it that documents originating from Christian, Jewish, and Magi sources all bear an almost identical phrase at the end of them? The only reasonable conclusion we can come to is that even though the covenants may be subject to some minor textual flaws, they all originate from the Prophet. As Agapius of Hierapolis, bishop of the north Syrian city of Manbij explained in his *Universal History*, which he wrote around 940 CE:

> "The Arabs mobilized at Yathrib. Head of them was a man called Muḥammad b. ʿAbdullāh and he became their chief and king ... Christians from among the Arabs as well as other people came to him. He granted them protection and wrote for them documents and he did so to all other nations who opposed him. By that I mean the Jews, Magi, Sabaeans and others. They gave him allegiance and took from him a guarantee of safety on the condition that they would pay him the jizya and kharāj." [63]

Similarly, the Coptic historian Jirjis b. al-ʿAmīd al-Makīn (d. 1273 CE) reported:

> "The Christians from among the Arabs and other [nations] came to him, so he granted them protection and wrote documents for them. He did the same with the Jews, the Magi, the Sabaeans as well as others, so they pledged allegiance to him and were granted protection in return for paying the jizya and kharāj[64] ... Christian chronicles report that he was benevolent and compassionate to them so they sent him a delegation requesting his protection. In return he imposed on them the jizya, was gracious to them, and wrote for them documents to guarantee their protection. He informed ʿUmar: 'Say to them that their livelihoods, wealth and honour is exactly the same as ours' ... He also said: 'Whoever oppresses a protected person he shall be his foe on the Day of Judgment (man ẓalama dhimmiyyan kāna khaṣmahu yawm al-qiyāma),' and: 'Whoever harms a protected person has harmed me (man adhā dhimmiyyan fa-qad ādhānī).'"
> 65

Logic dictates that these different documents which the Prophet issued would have borne a number of common phrases, features, and characteristics, and so it should come as no surprise to find the same stringent warning on the severity of harming a dhimmī at the end of them.

## 4. The Curious Silence in Muslim Tradition

As the authenticity of the Prophet's warning not to harm a dhimmī is attested in different covenants issued by him, it is important to point out that this same statement cannot be found in the main ḥadīth collections. Based on the documentary evidence, we know that this Prophetic saying began being uttered in 2 AH in Madīna and that it was continuously re-iterated until 9 AH. Though the recension of the *Covenant with the Children of Israel* does not recall any witnesses' names, the *Covenant with the Jews of Khaybar and Maqnā*, written in 9 AH, lists Abū Dharr as one of its witnesses.[66] Out of the 11 Companions whose names re-occur in the three Christian covenants in which we find this Prophetic utterance, seven of them can be found as witnesses to the *Covenant with the Magi*: ʿAlī b. Abī Ṭālib, Abū Dharr; Abū Bakr; ʿUmar b. al-Khaṭṭāb; ʿUthmān b. ʿAffān; Ṭalḥa b. ʿUbaydullāh; and Saʿd b. ʿUbāda. These seven Companions, among whom are the Rightly Guided Caliphs, would have been intimately

---

[62] See (El-Wakil 2017), "Searching for the Covenants."
[63] (Hoyland 2011, p. 87). For the Arabic, see (Mahboub De Mendbidj, known as Agapius of Hierapolis 1909, pp. 196–97); Hoyland's translation was edited by author.
[64] (al-Makīn 1625).
[65] al-Makīn, *ibid.*, p. 11.
[66] (Hirschfeld 1903), "The Arabic Portion of the Cairo Genizah at Cambridge," p. 172.

familiar with the ḥadīth and we ought, in theory, to find numerous solid chains of transmissions going back to them in the books of tradition. Despite the fact that ʿAlī was scribe to all covenants of the Prophet which include this stern warning at the end of them—with the exception of the *Covenant with the Copts* (and the Syriac Jacobites)—it can nevertheless not be found as a ḥadīth in Shīʿa sources.

As the covenants were intended to be valid until the Day of Judgment, it would make no sense to argue that the Prophet came to abrogate them before his death. The covenants of ʿAlī with the Magi[67] and with the Armenian Christians,[68] and the covenants of ʿUmar with the Christians of Jerusalem,[69] the Syriac Jacobites,[70] and the Christians of Mesopotamia[71] all follow the same tone and spirit as the Prophet's covenants, meaning that the Companions never regarded these precious documents as having been annulled.

Taking into account how the Prophet would have uttered the ḥadīth about harming a dhimmi pretty much from the time he moved to Madīna until the very end of his life, this legal maxim should be one of the most rigorously authenticated mutawātir traditions. Trying to understand why this is not the case is not so straightforward, particularly when we consider how what constitutes a mutawātir tradition is highly problematic. This was clearly noted by G.H.A. Juynboll when he explained that taking a report's recurrence at face value is deceiving, for the very definition of what is labelled "mutawātir" is itself an oxymoron:

> "For all canonical or non-canonical traditions, labelled *mutawātir* or otherwise, to be found in Muslim *ḥadīth* literature, not a single one has proto-wording supported by isnād strands which, when *analytically surveyed together*, show the requisite number of authorities—three, four, five or more—in *every* tier, i.e., on every separate level of transmission, from the very beginning to the very end.

> The only criterion that is found to apply to various so-called *mutawātir* transmissions is that of the requisite number of different transmitters in the *oldest* tier, i.e., the number of Companions allegedly transmitting one and the same saying from the Prophet or reporting on one and the same event in his life. But in later tiers of the *isnād* strands within these transmissions this requisite number cannot be established." [72]

Juynboll's observations, as Hüseyin Hansu has pointed out, were noted by ḥadīth masters such as Ibn Ḥibbān (died 965 CE/354 AH) who conceded to the fact "that all ḥadīths are āḥād."[73] Even traditions which re-occur in multiple isnāds show no pattern of organic growth. The confidence of Muslim scholars over the sanctity of mutawātir transmissions was, therefore, viewed with a great degree of skepticism by Ignaz Goldziher:

> "With pious intention, fabrications were combated with new fabrications, with new ḥadīths which were smuggled in and in which the invention of illegitimate ḥadīths were condemned by strong words uttered by the Prophet . . . The most widely spread polemical ḥadīth of this nature is the saying which survives in many versions: . . . 'Man who lies wilfully in regard to me enters his resting place in the fires of hell.' About eighty companions—not counting some paraphrases—hand down this saying, which is recognizable as a reaction against the increasing forgery of prophetic sayings." [74]

---

[67]  (Jejeebhoy 1851), *Tuqviuti-din-i-Mazdiasna*, pp. 12–18; *Kalima Ṭayyiba*, p. 64.
[68]  (Avdall 1870, p. 60).
[69]  (al-ʿĀrif 1999). Also see (Gabriel 1900), *Tārīkh al-Kanīsa*, pp. 585–87. (Akyüz 2002), *Osmanlı Devletinde Süryani Kilisesi*, pp. 146–147; Ottoman Archives in Istanbul, *The Church Registers* (*Kamame Kilisesi Defteri*), Register No. 8 (A.DVNS.KLS.d; 1171 AH), p. 5.
[70]  (Nau 1915, pp. 276–79). For a summary of this covenant from *the Life of Gabriel of Qartmin*, see (Hoyland 1997).
[71]  (Scher [1918] 1983), *Histoire Nestorienne Inédite: Chronique de Séert*, pp. 300, 620.
[72]  (Juynboll 2001, pp. 329–30).
[73]  (Hansu 2009), "Notes on the Term Mutawātir and its Reception in Ḥadīth Criticism," p. 395.
[74]  (Goldziher 1971).

The notion of a mass proliferation of ḥadīth forgeries was later elaborated by Joseph Schacht who expressed "that the great majority of traditions from the Prophet are documents not of the time to which they claim to belong, but of the successive stages of development of doctrines during the first centuries of Islam."[75] According to Schacht new isnāds kept on being reproduced to back a maxim whose intent was to alter an existing legal or social trend. Al-Nawāwī implicitly acknowledged this phenomenon in his introduction to his famous collection of 40 ḥadīths, *Al-Arbaʿīn*, whereby he explains that some texts kept on being reproduced and re-construed with slight variations using forged isnāds. As he observes, a report's multiplicity does not necessarily guarantee its authenticity:

> "It has been related to us from ʿAlī b. Abī Ṭālib, ʿAbdullāh b. Masʿūd, Muʿādh b. Jabal, Abū al-Dardāʾ, Ibn ʿUmar, Ibn ʿAbbās, Anas b. Mālik, Abū Ḥurayra, Abu Saʿīd al-Khudrī through many chains of transmission and in various forms that the Messenger of Allah said: 'Whoever from my umma has memorized 40 traditions relating to their religion will be raised by Allah on the Day of Judgment in the company of jurists and scholars'. In another narration he said: 'He will be raised as a jurist and scholar'. In the narration on the authority of Abū al-Dardāʾ he said: 'I will be on the Day of Judgment a witness and intercessor for him'. In a narration on the authority of Ibn Masʿūd he informs us that: 'It will be said to him enter paradise from whichever gate you wish'. In the narration of Ibn ʿUmar it is said: 'He will be classed as a scholar and raised among the martyrs'. According to their classification deriving from their numerous works, the scholars of ḥadīth have agreed that this is a weak tradition even though it has been transmitted by numerous chains of transmission." [76]

Schacht contended that the reproduction of texts went parallel with ascribing credible authorities to the newly created isnāds and that "Generally speaking, we can say that the most perfect and complete isnāds are the latest."[77] As the ḥadīths usually disseminate from a central authority before branching out into several strands, Schacht identified this central transmitter as the "Common Link"[78] while the term "Partial Common Link"—which was coined by Juynboll—referred to transmitters who relayed the narration from the Common Link to at least two of their own students.[79] Now that we have the covenants at hand, we will utilize them to better understand ḥadīth transmissions and cast new light on the ḥadīth corpus.

## 5. Harming a Dhimmī in the Books of Ḥadīth

The earliest ḥadīth recording how the Prophet will be the foe of whoever harms a dhimmī is in the *Kitāb al-Kharāj* of Abū Yūsuf (died 798 CE/181 AH), which informs us that:

> "It was narrated to me by some of those scholars who came before us (baʿd al-mashāyikh al-mutaqaddimīn) and who have raised this ḥadīth to the Prophet—peace and blessings be upon him—that when he appointed ʿAbdullāh b. Arqam to take the jizya from the protected people he said to him: 'Whoever oppresses a person with whom we have made a contract with (man ẓalama muʿāhidan), or places a burden on him more than he can bear, or takes away any of his rights, or takes something from him unwillingly (bi-ghayr ṭīb nafs), then I shall be his foe on the Day of Judgment (fa anā ḥajījuhu yawm al-qiyāma)." [80]

This narration seems to demonstrate intimate knowledge of the covenants. That nothing should be taken from the protected people unwillingly is reflective of the Prophet's Master Template with the

---

75  (Schacht 1953).
76  (al-Nawāwī 1984).
77  (Schacht 1953), *The Origins of Muhammadan Jurisprudence*, p. 165.
78  Schacht, *ibid.*, pp. 171–172.
79  (Juynboll 1983).
80  (Abū Yūsuf 1999).

Christian communities of his time, which states that the taxes levied on them should be by what "they willingly consent (bi-mā tuṭīb bihi anfusihim)."[81] As for the phrase that the Prophet shall be "his foe" (i.e., "ḥajījuhu") on the Day of Judgment, it is remarkably enough mirrored in the Prophet's *Covenant with the Children of Israel*. The same tradition was also reported by Ibn Zanjawayh (died 865 CE/251 AH) in his *Kitāb al-Amwāl* but with a complete isnād and a slight addition at the end:

> "Yūsuf b. Yaḥyā narrated to us from Ibn Wahb from Abū Ṣakhr al-Madanī that Ṣafwān b. Sulaym informed him on the authority of 30 children of the Companions of the Messenger of Allah—peace and blessings be upon him—from their fathers, directly from the Messenger of Allah—peace and blessing be upon him—that he said: 'Whoever oppresses a person with whom we have made a contract with (man ẓalama muʿāhidan), or places a burden on him more than he can bear, or takes away any of his rights, or takes something from him unwillingly (bi-ghayr ṭīb nafs minhu) then I shall be his foe on the Day of Judgment (fa anā ḥajījuhu yawm al-qiyāma)'. The Messenger of Allah–peace and blessings be upon him–then pointed to his chest with his fingers and said: 'Whoever kills a person who has made a contract with us and who has the protection of Allah and His messenger (lahu dhimmatu Allāhi wa rasūlihi), Allah has forbidden him the scent of paradise even though its scent can be felt at a distance of 70 years.'" [82]

This tradition was also relayed by al-Bayhaqī (died 1066 CE/458 AH) in his *Sunan al-Kubrā*. It was transmitted to him by two direct informants who both received the tradition from Abū al-ʿAbbās Muḥammad b. Yaʿqūb, from Muḥammad b. ʿAbd al-Ḥakam, from Ibn Wahb, from Abū Ṣakhr al-Madanī, from Ṣafwān b. Sulaym on the authority of 30 children of the Companions of the Messenger of Allah, from their fathers, directly from the Messenger of Allah.[83] The number 30 is certainly worthy of note because the covenants tend to have 30 witnesses to them. As for the prohibition in the above ḥadīth to kill any of the protected people who have the dhimma of Allah and His messenger, we now know that this can only be a reference to the covenants which all grant the non-Muslims the protection of Allah and His messenger.

Out of the six canonical books of tradition, we only find Abū Dāwūd (died 889 CE/275 AH) who narrated in his *Sunan* a tradition that is very similar to that of Ibn Zanjawayh and al-Bayhaqī and which al-Albānī classified as ṣaḥīḥ. According to Abū Dāwūd, the narration was transmitted to him from Sulaymān b. Dāwūd al-Mahrī, from Ibn Wahb, from Abū Ṣakhr al-Madanī, from Ṣafwān b. Sulaym on the authority of a number of children of the Companions of the Messenger of Allah, from their fathers, directly from the Messenger of Allah.[84] The isnāds provided by Abū Dāwūd, Ibn Zanjawayh, and al-Bayhaqī all render Ibn Wahb as the Common Link. Ibn Zanjawayh also narrates another similar tradition:

> "Yūsuf b. Yaḥyā narrated to us from Ibn Wahb from al-Ḥajjāj b. Ṣafwān al-Madīnī from Ibrāhīm b. ʿAbdullāh b. ʿAbd al-Raḥmān, from his father, that the Messenger of Allah—peace and blessings be upon him—said: 'Whoever oppresses a person with whom we have made a contract with (man ẓalama muʿāhidan) I shall be his foe on the Day of Judgment (fa anā ḥajījuhu yawm al-qiyāma); whoever sells a free person and consumes the money he has obtained from such a sale, I shall be his foe on the Day of Judgment (fa anā ḥajījuhu yawm al-qiyāma); and whoever is unjust to a person who is owed a wage I shall be his foe on the Day of Judgment (fa anā ḥajījuhu yawm al-qiyāma)'". [85]

---

[81] (El-Wakil and Nasrallah 2017) "The Prophet Muḥammad's Covenant with the Armenian Christians," sct. 33, pp. 495–96.
[82] (Ibn Zanjawayh 1986).
[83] (Abū Bakr Aḥmad b. Mūsā al-Bayhaqī 2003).
[84] (Abū Dāwūd 2009).
[85] (Ibn Zanjawayh 1986), *Kitāb al-Amwāl*, p. 381.

Another early authority to narrate this tradition but in a slightly abridged format was Yaḥyā b. Adam (died 818 CE/203 AH) in his *Kitāb al-Kharāj*:

> "Ibrāhīm b. Abī Yaḥyā narrated to us from al-ʿAbbās b. ʿAbd al-Raḥmān from Zayd b. Rufayʿ that the Messenger of Allah–peace and blessings be upon him–said: 'Whoever oppresses a person with whom we have made a contract with (man ẓalama muʿāhidan), or places a burden on him more than he can bear, then I shall be his foe until the Day of Judgment (fa anā ḥajījuhu ilā yawm al-qiyāma)'". [86]

Abū Nuʿaym al-Iṣfahāni (d. 1038 CE/430 AH) narrates another variant in which the Prophet's injunction is justified because the non-Muslims are living in a state of humiliation, having accepted payment of the jizya:

> "Muḥammad b. Ḥumayd narrated to us from ʿUmar b. al-Ḥasan al-Qāḍī al-Ḥalabī from Ayūb al-Wazān from Yaʿlā b. al-Ashdaq from ʿAbdullāh b. Jarād that the Messenger of Allah—peace and blessings be upon him—said: 'Whoever oppresses a dhimmī (man ẓalama dhimmiyyan) who is paying the jizya and who accepts being humiliated, then I shall be his foe on the Day of Judgment (fa anā khaṣmuhu yawm al-qiyāma)'". [87]

Al-Balādhurī (dated 892 CE/279 AH) also reports an account which denotes awareness of the ḥadīth:

> "Muḥammad b. Saʿd narrated from al-Wāqidī: Some people in Lebanon rebelled because they were complaining about the collector of the *kharāj* in Baʿalbek. This made Ṣāliḥ b. ʿAlī b. ʿAbdullāh b. ʿAbbās send troops against them to destroy their fighting power and to allow the rest of the population to retain their [Christian] faith. Ṣāliḥ sent them back to their villages but expelled other natives of Lebanon. Al-Qāsim b. Sallām related to me on the authority of Muḥammad b. Kathīr that Ṣāliḥ received a long communication from al-Awzāʿī of which the following extract has been preserved: 'You have heard of the expulsion of the protected people from Mount Lebanon although they did not side with those who rebelled—many of whom you killed and the rest which you allowed to return to their villages. How then can you punish the many for the fault of the few and make them leave their homes and possessions in spite of Allah's decree that 'no soul shall bear the burden of another (Q6: 164)'. The most rightful course of action for you is to abide and obey the command of the Prophet with the strictest of observance when he said 'Whoever oppresses a person with whom we have made a contract with (man ẓalama muʿāhidan), or places a burden on him more than he can bear, I shall be his foe (fa anā ḥajījuhu)." [88]

ʿAlī Ḥasan al-Jalabī records in his *Mawsūʿat al-Aḥādīth wa al-Athār al-Ḍaʿīfa wa al-Mawḍūʿa* eight variants of the ḥadīth "man adhā dhimmiyyan". They are:

1.  Whoever harms a protected person I am his foe (man adhā dhimmiyyan fa anā khaṣmuhu);
2.  Whoever harms a protected person I am his foe, and whoever I am his foe then I shall be his foe on the Day of Judgment (man adhā dhimmiyyan fa anā khaṣmuhu wa man kuntu khaṣmahu khāṣamtuhu yawm al-qiyāma);
3.  Whoever harms a protected person I am his foe on the Day of Judgment (man adhā dhimmiyyan fa anā khaṣmuhu yawm al-qiyāma);
4.  Whoever harms a protected person has harmed me (man adhā dhimmiyyan fa-qad ādhānī);

---

[86] (Adam 1987).
[87] (Abū Nuʿaym 1998).
[88] (Aḥmad b. Yaḥyā al-Balādhurī 1987). My translation s relied on that of Hitti. See (Aḥmad b. Yaḥyā al-Balādhurī 2011). Also see (Ibn Zanjawayh 1986), *Kitāb al-Amwāl*, ḥadīth No. 689, p. 420.

5.     Whoever harms a protected person I am his foe (man adhā dhimmiyyan kuntu khaṣmahu);

6.     Whoever harms a protected person I shall be his foe on the Day of Judgment (man adhā dhimmiyyan kuntu khaṣmahu yawm al-qiyāma);

7.     Whoever oppresses a protected person Allah shall be his foe on the Day of Judgment or I shall be his foe (man ẓalama dhimmiyyan kān Allāhu khaṣmahu yawm al-qiyāma aw kuntu khaṣmahu);[89]

8.     Whoever maligns a protected person I am his foe (man adhmā dhimmiyyan fa anā khaṣmuhu)."[90]

Al-Khaṭīb al-Baghdādī produces a full isnād for variant number 2 on the authority of the Companion 'Abdullāh b. Mas'ūd. He then explains how this ḥadīth is rejected (munkar) due to the weakness of one of the narrators.[91] Ibn al-Qayyim (died 1350 CE/751 AH) showed awareness of the ḥadīth by quoting al-Khaṭīb (though he mistakenly attributes the ḥadīth to the Companion Jābir b. 'Abdullāh) after which he narrates a report on the authority of Aḥmad b. Ḥanbal that this ḥadīth is one of four popular ḥadīths that are propagated on the authority of the Prophet but which have no foundation (laysa laha aṣl).[92] The tradition which Aḥmad was aware of is variant number 4 which can be found in its original version in the covenants with the Children of Israel and with the Magi, both of which carry the statement "man ādhāhum ādhanī". Al-Suyūṭī (died 1505 CE/911 AH), elaborating on Ibn al-Qayyim states that according to Abū al-Faḍl al-'Irāqī (died 1403 CE/806 AH), it was impossible for Aḥmad b. Ḥanbal to have stated that the ḥadīth is one of four rejected traditions.[93] Al-Suyūṭī then quotes the tradition from Abū Dāwūd explaining that its isnād is sound, and even though the names of the Companions are not mentioned, al-Suyūṭī is the only scholar to contend that it has reached the level of tawātur.[94] Al-Shawkānī (died 1834 CE/1250 AH) classified the tradition as baseless (mawḍū'), though he notes how al-'Irāqī stated that it has its chains of transmission (lahu ṭuruq).[95] Ibn Taymiyya took a more extreme position, stating: "This is a lie attributed to the Prophet, peace and blessings be upon him. No one of the people of knowledge has narrated it."[96] He then proceeds to explain how Muslims should discriminate against non-Muslims but not to treat them unjustly. According to al-Albānī, variant 4 of the ḥadīth is weak and has no foundation. He explains that neither al-Ṭabarānī reported it in *al-Awsaṭ* nor anybody else. He then states that the correct wording of the ḥadīth is, in fact, "*m*an ādhā musliman"—"whoever harms a Muslim".[97] It may be of interest to point out here that another variant recorded by al-Jalabī, and which is similar in spirit to the tradition being studied, is "Whoever injures a protected person shall be punished/lashed on the Day of Judgment with whips made of fire."[98]

It is highly significant that though the ḥadīth was known to Muslim scholars, it was at no point considered authoritative, only marginally accepted at best. Despite the fact that the ḥadīth has three complete isnāds, one going back to Ibn Mas'ūd, another to 'Abdullāh b. Jarād, and a third to Zayd b. Rufay', and that it has the transmitter Ibn Wahb as an established Common Link, it did not at any point embody a pattern of organic growth with recurrent isnāds being visible at every tier of transmission and going back to well-known Companions of the Prophet. The ḥadīth also seems to have been subject to slight modifications to suit the cultural context of the time. So, for instance, we see how after narrating variant 7 of the ḥadīth, Ibn Taymiyya states "This is weak, but the narration that is correct is 'Whoever kills a person with whom a contract has been established without any due right,

---

[89]   For these seven variants see Alī Ḥasan al-Jalabī's work (al-Jalabī 1999).
[90]   al-Jalabī, *ibid.*, p. 191. It appears that the origin of this tradition lies with the word "adhā" having been misread as "adhmā".
[91]   (al-Baghdādī 2001).
[92]   (al-Jawzī 1966).
[93]   ('Abd al-Raḥmān Jalāl al-Dīn al-Suyūṭī 1975).
[94]   al-Suyūṭī, *ibid.*, p. 141.
[95]   (al-Shawkānī 1995).
[96]   (Ibn Taymiyya 2004). Also see (Ibn Taymiyya n.d.).
[97]   (al-Albānī 1980).
[98]   (al-Jalabī 1999), *Mawsū'at al-Aḥādīth wa al-Athār al-Ḍa'īfa wa al-Mawḍū'a*, vols. 10, p. 183.

then such a person will not smell the scent of paradise.'"[99] For intricate legal reasons, the narration about oppressing a person with whom a contract has been made (i.e., muʿāhidan) seems to have been more palatable to Ibn Taymiyya than its variant concerning the oppression of a protected person (i.e., dhimmī) who was a subject of the Islamic state.

## 6. Textual Analysis in the Context of Multiple Transmissions

It is certainly perplexing how on no occasion in the Islamic sources do we find an identical or close recension of the *Covenant with the Christians of Najran* that exists in the *Chronicle of Seert*. The *Najran Compact*[100] in the Islamic historiographical works is either a document that was granted to both Najran's Christian and Jewish populations or a composite text that was based on an amnesty given to the Jews of Najran.[101] Though the trustworthiness of a report requires analysis of both its text (matn) and isnād, it should be here pointed out that the isnād alone does not prove anything. As the *Najran Compact* is a defective text which has been transmitted through numerous chains of transmission, a more conducive approach for scholars may, therefore, require placing greater emphasis on textual analysis rather than on the multiplicity and strength of the isnāds to determine the extent of a report's veracity.

The *Kitāb al-Kharāj* of Abū Yūsuf is the earliest work to transmit the *Najran Compact*, and like the *Kitāb al-Siyar al-Ṣaghīr* of al-Shaybanī (died 805 CE/189 AH)[102] and *al-Ṭabaqāt al-Kubrā* of Ibn Saʿd (died 845 CE/230 AH),[103] it does so without relaying an isnād. Nevertheless, it is possible to locate 12 separate chains of transmission for the *Najran Compact* in the books of tradition. Though none of these isnāds are of the highest caliber (ṣaḥīḥ), they do, nevertheless, all bear the characteristics one would expect to find of typical ḥadīth transmissions. The transmitter ʿUbaydullāh b. Abī Humayd is a Common Link, and the narrator ʿĪsā b. Yūnus a Partial Common Link in the first isnād bundle (Appendix A, section (1)). The second isnād bundle traces the *Najran Compact* to Ibn ʿAbbās through a single-stranded isnād and through two chains of transmission that have as their Common Link Yūnus b. Bukayr (Appendix B, section (2)). Ibn Zanjawayh reports one chain through Yaʿlā b. ʿAbīd[104] and another through Muḥaḍir b. al-Muwarriʿ,[105] both having al-Aʿmash as the Common Link and going back to the authority of Sālim b. Abī al-Jaʿd. Apart from these nine transmissions, we find three independent single-stranded transmissions in the books of tradition which either reproduce or reference the *Najran Compact*:

1. Abū ʿUbayd > ʿUthmān b. Ṣāliḥ > Ibn Lahīʿa > Yatīm ʿUrwa > ʿUrwa b. al-Zubayr;[106]
2. Bakr b. al-Haythamī > ʿAbdullāh b. Ṣāliḥ > al-Layth b. Saʿd > Yūnus b. Yazīd al-Aylī > al-Zuhrī;[107]
3. Husayn b. ʿAlī al-Aswad > Yaḥya b. Adam > unknown > Ḥasan b. Ṣāliḥ.[108]

The recension recorded by Abū Yūsuf is said to have been written by ʿAbdullāh b. Abī Bakr and it has five witnesses' names at the end of it, meaning that it was most likely derived from an original document, though we cannot vouchsafe the extent of its textual integrity.[109] Documents

---

[99] (Ibn Taymiyya 2004), *Majmuʿat Fatāwī*, vol. 18, p. 128.

[100] Throughout this article, the term "Compact" is used to refer to documents emerging from Muslim sources, while the term "Covenant" is employed in reference to documents originating from non-Muslim sources.

[101] (El-Wakil 2017) "Searching for the Covenants," pp. 40–48.

[102] (Abū ʿAbdullāh Muḥammad b. al-Ḥasanal-Shaybānī 1975).

[103] (Ibn Saʿd 2001).

[104] (Ibn Zanjawayh 1986), *Kitāb al-Amwāl*, ḥadīth No. 418, p. 276.

[105] Ibn Zanjawayh, *ibid.*, ḥadīth No. 419, p. 277.

[106] Ibn Zanjawayh, *ibid.*, ḥadīth No. 733, pp. 451–52.

[107] (Aḥmad b. Yaḥyā al-Balādhurī 1987), *Kitāb Futūḥ al-Buldān*, p. 85. Also see (Aḥmad b. Yaḥyā al-Balādhurī 2011), *The Origins of the Islamic State*, p. 98.

[108] al-Balādhurī, *ibid.*, pp. 86–87. Also see (Aḥmad b. Yaḥyā al-Balādhurī 2011), *The Origins of the Islamic State*, p. 99.

[109] (Abū Yūsuf 1999), *Kitāb al-Kharāj*, pp. 84–85.

written to the people of Najran by 'Alī[110] and al-Mughīra b. Shu'ba[111] have also been reported in the historiographical works, which potentially means that more than one document was granted to them.

　　Though the *Covenant with the Christians of Najran* is not dated, it shares a common anomaly with the remaining Prophetic covenants by having Mu'āwiya as its scribe. One recension of the *Covenant with the Monks of Mount Sinai* has Mu'āwiya in the list of witnesses[112], and his name as scribe to the *Covenant with the Christians of the World* written on Monday 29 Rabī' al-Thānī 4 AH and to the *Covenant with the Armenian Christians* written on a Monday in Dhū al-Ḥijja 2 AH[113] points to an earlier date in which he embraced Islam. The Islamic sources only record five correspondences which Mu'āwiya wrote on behalf of the Prophet, with only one of them being dated, i.e., the *Compact with al-'Alā' b. al-Ḥaḍramī*, which was written on 3 Dhū al-Qa'da 4 AH.[114]

　　The fact that Mu'āwiya's prominent role as scribe of the revelation is virtually non-existent in the books of tradition can only lead us to conclude that a number of sacred texts were redacted in the 'Abbasīd era. This is supported by Goldziher's observation that when al-Ma'mūn acceded to the Caliphate, he made sure to send an announcer to the streets to declare that the Caliph would not extend his protection to anyone who would mention Mu'āwiya favorably.[115] Even so, Muslim historical recollection could not completely do away with Mu'āwiya's important status during early Islam. Though there remains much controversy about his character, even the faith upon which he died, he seems to have had at one point a fair number of advocates. We, therefore, find in *Tārikh Baghdād* a report attributed to al-Ma'āfī b. 'Imrān vehemently defending Mu'āwiya by stating that he was the Companion, brother-in-law, scribe, and trustee of the Prophet who wrote down the revelation.[116] A number of weak ḥadīths also attest to this: one tradition attributed to the Prophet states "Trustworthiness in the sight of Allah is with three: Myself, Gabriel, and Mu'āwiya."[117] Another variant reads: "Allah has secured his revelation with three: Gabriel in the Heavens, Muḥammad on the earth, and Mu'āwiya b. Abī Sufyān."[118] It is even said that Gabriel came to the Prophet when Mu'āwiya was in the middle of writing and informed him: "O Muḥammad! Verily, your scribe is trustworthy!"[119] and that "Allah revealed to the Prophet—peace and blessings be upon him: Have Mu'āwiya as your scribe for he is trustworthy and can be relied upon (amīn, ma'mūn)."[120] One fantastic tradition even goes so far as to state that Gabriel gave the Prophet a pen that descended from the divine throne and which he gifted to Mu'āwiya to write *ayat al-kursī*![121] His role as scribe was so well known that even in *The Disputation of the Monk Abraham of Tiberias* he is referred to as "the scribe of the revelation."[122]

　　Our hypothesis of sacred texts having been redacted during the 'Abbasīd era can further be tested by turning our attention to the traditional account of the compilation of the Qur'ān, which was allegedly concocted by the Common Link al-Zuhrī to alienate 'Alī from the compilation process.[123] If, indeed, Mu'āwiya was a scribe of the Prophet who had written many letters and covenants on the latter's behalf, why would al-Zuhrī—as forger of the traditional account living in the era of the Umayyads—have omitted Mu'āwiya's name? As 'Uthmān's most trusted governor, it would have made sense to at least include the Prophet's scribe as part of the committee that assisted the third

---

[110] (Aḥmad b. Yaḥyā al-Balādhurī 2011), *The Origins of the Islamic State*, p. 101.
[111] (Hamidullah 1987, p. 179).
[112] (Library of Congress MS 695).
[113] For a discussion of these anachronisms and the case for Mu'āwiya's early conversion, see (El-Wakil 2016), "The Prophet's Treaty with the Christians of Najran," pp. 286–92.
[114] For an in-depth discussion, see (El-Wakil 2019).
[115] (Goldziher 1971), *Muslim Studies*, vol. 2, p. 54.
[116] (al-Baghdādī 2001), *Tārīkh Baghdād*, vol. 1, p. 577.
[117] (Ibn 'Irāqa 1981).
[118] Ibn 'Irāqa, *ibid.*, vol. 6, p. 4.
[119] Ibn 'Irāqa, *ibid.*, vol. 5, p. 4.
[120] Ibn 'Irāqa, *ibid.*, p. 4.
[121] Ibn 'Irāqa, *ibid.*, vol. 3, p. 4.
[122] (Szilágy 2014).
[123] See (El-Wakil 2015).

Caliph in the compilation of the Islamic scripture. Even if we are to assume that Muʿāwiya was not involved in any way in the collection of the Qurʾān, giving him a fictitious or limited role in such a sacred endeavor would not only have been acceptable to the Umayyads and their subjects, but it would also have been a credible historical fabrication. If al-Zuhrī wanted to concoct a tale about the collection of the Qurʾān for political expediency, then having Muʿāwiya assisting the three Rightly Guided Caliphs would have been a golden opportunity to gain favor from his political masters. Curiously enough, neither he nor ʿAlī are mentioned in the traditional account even though both these men were, according to the covenants, the Prophet's most prolific scribes.

All in all, it is certainly very suspicious that on no occasion in the most trusted books of tradition we find scholars from the Umayyad period transmitting an official decree scribed by the founder of the dynasty in which they were living. It also makes no sense for Christians to have inserted Muʿāwiya's name as scribe to forged covenants after the fall of the ʿUmayyads. In light of these inconsistencies, the use of multiple chains of transmission and the attribution of a tradition to a Common Link becomes highly questionable, and we may here argue that the traditional account of the Qurʾān's compilation attributed to al-Zuhrī was most likely a deliberate attempt to ascribe a historical fiction to a well-known authority. In the case of the *Najran Compact,* the multiple isnāds and the Common Links that we find do not exert a pattern of reliable transmission, but just like the traditional account of the Qurʾān's compilation, it showcases that the isnād was, at times, used as a tool for reproducing a particular text by ascribing it to disparate authorities, among them a so-called "Common Link".

## 7. Are the Ḥadīths in Harmony with the Covenants?

Schacht argued that the backward growth of isnāds was "identical with the projection of doctrines to higher authorities"[124] and that "We often find that traditions are formulated polemically with a view to rebutting a contrary doctrine or practice."[125] In order to assess Schacht's theories and to determine to what extent the ḥadīths are reflective of the true teachings of the Prophet and the Rightly Guided Caliphs, we will here compare the text and isnād of a number ḥadīths to the covenants. Six examples will be listed which, if Muslim scholars can reconcile with the covenants, would mean that we are dealing with "seemingontradictions". If, however, they are unable to do so, then the examples listed below would validate Schacht's theories (with regards to the rights of non-Muslims at least).

### 7.1. Granting the Protection of Allah and His Messenger

The covenants command the Muslims to give the protected people the protection of Allah and His messenger (dhimmatu Allāhi wa rasūlihi)[126], but the *Kitāb al-Athār* of Abū Ḥanīfa (died 767 CE/150 AH) discourages the Muslims from doing so, stating that "If they want you to give them the protection of Allah, do not give it to them, but give them yours and your fathers' protection instead for it is better that you violate the protection that you give to them rather than you violate the protection of Allah, the Mighty and Majestic."[127] Muḥammad b. Ḥasan al-Shaybānī provided the following strong isnād for this ḥadīth: Abū Ḥanīfa–ʿAlqama b. Marthad–Sulaymān b. Burayda—his father the Companion Burayda b. al-Ḥuṣayb al-Aslamī. A variant tradition in Muslim's *Ṣaḥīḥ*[128], the Sunan of

---

[124] (Schacht 1953), *The Origins of Muhammadan Jurisprudence*, p. 165.

[125] Schacht, *ibid.*, p. 152.

[126] To find this expression in various covenants, see (El-Wakil 2017), "Searching for the Covenants," p. 102, sct. 8 for the Covenant with the Banū Zakān; p. 106, sct. 8 for the Covenant with the Jews of Khaybar and Maqnā; 124, sct. 6 for the Covenant with the Magi; p. 131 sct. 4 for the Covenant of ʿAlī with the Magi. Also see (El-Wakil and Nasrallah 2017), "The Prophet Muḥammad's Covenant with the Armenian Christians," p. 504, sct. 54, where the concept of the protection of Allah and His messenger can be found.

[127] (Ash-Shaybani 2006). Translation was edited by author. Also see (Abū Nuʿaym 1994) and (Abū ʿAbdullāh Muḥammad b. al-Ḥasanal-Shaybānī 1975), *Kitāb al-Siyar al-Ṣaghīr*, p. 93.

[128] (Muslim 2006).

Abū Dāwūd,[129] and the *Jāmiʿ* of al-Tirmidhī[130] point to ʿAlqama b. Marthad as the Common Link. Seemingly independent yet similar traditions can be found in the Musnad of Zayd b. ʿAlī on the authority of his father Imām Zayn al-ʿĀbidīn—from al-Ḥusayn—from ʿAlī b. Abī Ṭālib[131] as well as in al-Kulaynī's *al-Kāfī* on the authority of Imām Jaʿfar al-Ṣādiq.[132] We know the concept of granting the protection of Allah and His messenger to be historically factual, not only based on the Jerusalem 32 inscription, but also because of two letters written in the 680s CE which were discovered in the village of Nessana in Palestine.[133] It is therefore impossible for any trustworthy Companion or any members of the Prophet's household to have discouraged Muslims from granting the protection of Allah and His messenger to non-Muslims when the Prophet and ʿAlī had themselves done so.

*7.2. Taxation*

The Prophet's Master Template with the Christian communities of his time stipulates that religious authorities would be exempt from all taxes, though it requires ordinary folk to pay 4 dirhams as poll-tax and land-owners to pay 12 dirhams as kharāj.[134] It was usually accepted for women to be exempt from the poll-tax but not from the kharāj[135], though some scholars debated whether the poll-tax should also be imposed on women and slaves. The different correspondences with the people of Yemen that have come down to us and which stipulate 1 dīnār as payment of the jizya are not consistent in that regard.[136] The Prophet's *Covenant with the Children of Israel* stipulates one and a half qafla on the poor and five on the rich,[137] implying that the jizya was only to be levied on free men.

The Ismāʿīlī scholar al-Qāḍī al-Nuʿmān (died 974 CE/363 AH) reports a tradition on the authority of the Prophet that whoever does not levy the jizya on a non-Muslim or intercedes on his behalf so that it not be imposed on him has betrayed Allah, the Prophet, and all of the Believers.[138] This is contradicted by the Prophet and ʿAlī's covenants with the Magi, the Prophet's *Covenant with the Jews of Khaybar and Maqnā*, and the *Covenant of ʿUmar with the Christians of Jerusalem*, which either stipulate exemption of the jizya or payment of it to the non-Muslims' religious authorities. Al-Qāḍī al-Nuʿmān concedes that if the non-Muslims participate in the military expeditions, then they should be recipients of a deduction in tax rates, but at no point does he endorse exemption from the jizya. He relays on the authority of ʿAlī how the jizya should only be levied on the non-Muslims' free men and not on their women, children, and slaves. Though such a stipulation no doubt conforms to the covenants, al-Qāḍī al-Nuʿmān proceeds to inform us that the amount they should pay is 48 dirhams per year for the upper classes, 24 dirhams for the middle classes, and 12 dirhams for the lower classes.[139] Al-ʿĀmilī tells us that it was ʿUmar who enforced these rates, but that he only did so after having consulted ʿAlī.[140]

A tradition with the reliable isnād of Mālik–Nāfiʿ-Aslam, the freed-slave of ʿUmar, tells us that ʿUmar b. al-Khaṭṭāb "imposed a jizya tax of 4 dīnārs on those living where gold was the currency, and 40 dirhams on those living where silver was the currency. In addition, they had to provide for the

---

[129] (Ibn Maja 1432 AH).
[130] (al-Tirmidhī 1996).
[131] (Zayd b. ʿAlī 1999).
[132] (Abū Jaʿfar Muḥammad b. Yaʿqūb al-Kulaynī 2000).
[133] See (Hoyland 2015).
[134] (El-Wakil and Nasrallah 2017), "The Prophet Muḥammad's Covenant with the Armenian Christians," pp. 473–74. For Arabic, see sct. 32, pp. 33, 495–96. For the tax stipulation of 4 dirhams which was levied on the Samaritans, see Abulfathi (al-Sāmirī 1865).
[135] (Tritton 1930, p. 198).
[136] (Hamidullah 1987), *Majmūʿat al-Wathāʾiq al-Siyāsiyya*, pp. 209, 221. The Prophet's letter to his governors in Yemen is, however, generic, stating that the jizya should be levied on every adult without specifying whether these should be male or female, free people or slaves (p. 201). The recension of the letter to Muʿādh b. Jabal states that it should be levied on every adult male and female (p. 213), while the second recension merely states on every adult (p. 214).
[137] (Ahroni 1998), "Some Yemenite Jewish Attitudes towards Muḥammad's Prophethood," vol. 97, pp. 84–85.
[138] (al-Qāḍī al-Nuʿmān 1963).
[139] al-Qāḍī al-Nuʿmān, *ibid.*, pp. 380–81.
[140] (Al-ʿĀmilī 2000).

Muslims and receive them as guests for three days."[141] The stipulation in the Master Template[142] to have the non-Muslims lodge the Muslims for three days was conveniently remembered outside the confines of its due context. The tax rates that ʿUmar supposedly imposed are in complete contradiction to the covenants. ʿUmar's Covenant with the Syriac Jacobites states that the jizya should be 4 dirhams, thereby implying that he followed the same policy as the Prophet with regards to taxation.[143] The only place where we see this tax stipulation in accordance with the covenants is in an account that can be found in al-Ṭabarī in which he reports that the Muslims took 4 dirhams from Bārūsmā, Nahr Jawbar and al-Zawābī.[144]

The covenants prohibit the Muslims from taking the tithe (al-ʿushr) from the protected people,[145] yet numerous narrations bearing a strong isnād permit this. One narration in the Muwaṭṭaʾ bearing the isnād Mālik–al-Zuhrī–Sālim–his father ʿAbdullāh b. ʿUmar tells us that ʿUmar b. al-Khaṭṭāb would take a tenth of the pulses from the Nabatean Christians.[146] Another tradition has al-Zuhrī comment that al-Sāʾib b. Yazīd told him that "As a young man I used to work with ʿAbdullāh b. ʿUtba b. Masʿūd in the market of Madīna in the time of ʿUmar b. al-Khaṭṭāb and we used to take a tenth from the Nabateans."[147] When Mālik asked al-Zuhrī why ʿUmar did this, he replied that "It used to be taken from them in the jāhiliyya, and ʿUmar imposed it on them."[148] Mālik also heard that ʿUmar b. ʿAbd al-ʿAzīz (died 720 CE/101 AH) wrote to his governors stipulating that the protected people who trade in Muslim lands must pay a tenth "from what they invest in such trade."[149] ʿUmar b. ʿAbd al-ʿAzīz had allegedly followed the sunna of ʿUmar, but it is very clear from the conciliatory taxes found in ʿUmar's covenants with the Christians of Jerusalem and with the Syriac Jacobites that he did not impose the tenth, or if he ever did, then it would have been due to some particular circumstances. Even if we are to assume that there is some historical truth to ʿUmar's tax stipulations in the books of tradition, it is somewhat curious that the original provisions of 4 and 12 dirhams can never be found in any of the Muslim legal texts.

*7.3. Building of New Churches*

Abū ʿUbayd (died 839 CE/224 AH)[150] and Ibn Zanjawayh[151] report a tradition on the authority of the Prophet from "Ḥumayd, from al-Layth b. Saʿd, from Tawba b. Namir al-Haḍramī from those who told him that the Prophet said 'There is no razing [of old churches] in Islam and no building of new ones.'"[152] Al-Qāḍī al-Nuʿmān reports a tradition on the authority of ʿAlī from the Prophet prohibiting Muslims from entering churches because God's curse descends there (in contradiction to Q22:40) after which he narrates how the Prophet prohibited the building of new churches in Islamic lands.[153] Abū al-Shaykh al-Iṣfahānī[154] and Abū Nuʿaym[155] report a tradition with a complete isnād in which ʿUmar b. al-Khaṭṭāb heard the Prophet say "There is no new church in Islam and no parts of a

---

[141] (Mālik 1985). Also see (Ibn Zanjawayh 1986), *Kitāb al-Amwāl*, ḥadīth No. 593, pp. 368–69.
[142] (El-Wakil and Nasrallah 2017), "The Prophet Muḥammad's Covenant with the Armenian Christians," sct. 52, p. 503.
[143] (Nau 1915, pp. 276–79).
[144] (Muḥammad b. Jarīr al-Ṭabarī 1879).
[145] (El-Wakil 2017), "Searching for the Covenants," pp. 74–75.
[146] (Mālik 1985), *Muwaṭṭaʾ*, vol. 2, Book 17, ḥadīth No. 46, p. 281.
[147] Ibid., ḥadīth No. 47, p. 281.
[148] Ibid., ḥadīth No. 48, p. 281.
[149] Ibid., ḥadīth No. 45, p. 280.
[150] (Abū ʿUbayd 1989).
[151] (Ibn Zanjawayh 1986), *Kitāb al-Amwāl*, ḥadīth No. 398, p. 269.
[152] The Arabic word "kanīsa" can mean either a church or a synagogue. For the sake of simplicity, however, it has been translated as "church" throughout this paper.
[153] (al-Qāḍī al-Nuʿmān 1963), *Daʿāʾim al-Islām*, vol. 1, p. 381.
[154] (al-Iṣfahānī 1992), *Ṭabaqāt al-Muḥadithīn bi-Iṣfahān*, vol. 3, ḥadīth No. 355, p. 38.
[155] (Abū Nuʿaym 1990), *Dhikr Akhbār Iṣfahān*, vol. 1, p. 430.

church are to be renewed." Abū ʿUbayd[156] and Ibn Zanjawayh[157] report yet another tradition with an isnād going back to ʿUmar b. al-Khaṭṭāb that "There is no place for [new] churches in Islam nor for [old ones] to be razed." Though these traditions can be traced to the Prophet, they include the name of ʿUmar in their isnād who is credited more so than the Prophet for discriminatory practices against non-Muslims. Ibn Zabr (died 940 CE/329 AH) narrates two versions of the *Pact of ʿUmar* in his *Juzʾ fīhi Shurūṭ al-Naṣārā*, each one listing the prohibition of building new churches and both being preceded by a complete isnād.[158] Though we have traditions attributed either to the Prophet, ʿUmar, or ʿAlī justifying discriminatory practices against non-Muslims and the prohibition of building new churches, based on what we know of the covenants, all of these traditions are false.[159] These views appear to be reflective of the policy of later Caliphs, and this is evident by how ʿUmar b. ʿAbd al-ʿAzīz, in one of his letters whose contents have been preserved by Abū ʿUbayd[160] and Ibn Zanjawayh,[161] explained: "Do not destroy a synagogue, church, or fire temple, but also do not build new ones."

*7.4. Levying the Jizya on the Magi*

A peculiar tradition in the Muwaṭṭaʾ which contradicts the Prophet and ʿAlī's covenants with the Magi reads:

> "Yaḥya related to me from Mālik from Jaʿfar b. Muḥammad b. ʿAlī, from his father [i.e., most likely a reference to al-Ḥusayn as the source of the narration] that the matter of the Magi was mentioned to ʿUmar b. al-Khaṭṭāb for which he said 'I do not know how to deal with them'. ʿAbd al-Raḥmān b. ʿAwf then said, "I bear witness that I heard the Messenger of Allah—peace and blessings be upon him—say 'Deal with them as you do with the People of the Book.'" [162]

As ʿUmar was a witness to the Prophet's *Covenant with the Magi*,[163] it seems highly implausible that he would have been ignorant of how to deal with them during his Caliphate. The isnād for the above tradition is one of the strongest that exists but it is not possible for Imāms al-Ṣādiq, al-Bāqir, Zayn al-ʿAbidīn, and al-Ḥusayn, whose veracity is irreproachable, to have narrated the above tradition. ʿAbd al-Razzāq (died 827 CE/211 AH) reported the same ḥadīth from Ibn Jurayj, who also allegedly heard it from Imām al-Ṣādiq rendering the latter as a Common Link to something that is patently false.[164] It is also odd for Imām al-Ṣādiq not to have referred to the covenants of the Prophet or of ʿAlī, especially as ʿAlī was the scribe of the Prophet's *Covenant with the Magi* while al-Ḥusayn the scribe to ʿAlī's covenant with them.[165] Ibn Jurayj also reports that he heard the same ḥadīth through another chain of transmission from ʿAmr b. Dinār from Bajāla al-Tamīmī from ʿUmar b. al-Khaṭṭāb.[166] Sufyān b. ʿUyayna also reports having heard the tradition from ʿAmr b. Dinār rendering the latter as a Common Link.[167] As it is highly peculiar for ʿUmar not to have been aware of how the Prophet had levied the

---

[156]  Abū ʿUbayd, *Kitāb al-Amwāl*, ḥadīth No. 260, p. 176.

[157]  (Ibn Zanjawayh 1986), *Kitāb al-Amwāl*, ḥadīth No. 399, p. 269.

[158]  (Ibn Zabr 2006). For the first version of the Pact of ʿUmar, see pp. 22–23; for the second version, pp. 23–25. Three different versions with a full isnād can be found in Ibn ʿAsākir (1995).

[159]  It is significant that Muʿāwiya commanded that the Great Church at Edessa be rebuilt after it collapsed due the fact of an earthquake, most probably having done so based on the covenants of the Prophet which he had once scribed. See (Hoyland 2011, pp. 170–71).

[160]  (Abū ʿUbayd 1989), *Kitāb al-Amwāl*, ḥadīth No. 262, pp. 176–77.

[161]  (Ibn Zanjawayh 1986), *Kitāb al-Amwāl*, ḥadīth No. 400, pp. 269–70.

[162]  (Mālik 1985), *Muwaṭṭaʾ*, vol. 2, Book 17, ḥadīth No. 42, p. 278.

[163]  (El-Wakil 2017), "Searching for the Covenants," p. 128, sct. 14.

[164]  (al-Ṣanʿanī 1983). Also see vol. 6, pp. 69, 10025.

[165]  (El-Wakil 2017), "Searching for the Covenants," p. 127, sct. 12, 13, 135.

[166]  (al-Ṣanʿanī 1983), *Muṣannaf*, vol. 10, ḥadīth No. 19390, p. 367 and ḥadīth No. 19261, p. 327. Also see vol. 6, ḥadīth No. 9972, p. 49.

[167]  (Ibn Ḥanbal 2001); (al-Bukharī 2002); Abū Dāwūd, *Sunan*, vol. 4, ḥadīth No. 3043, p. 650.

jizya from the Magi until he was told by 'Abd al-Raḥmān b. 'Awf, we would have to discredit the above ḥadīth as a fabrication, especially its alleged provenance from the Prophet's family.

*7.5. Expelling the Jews and Christians from the Arabian Peninsula*

The terms and conditions of the Prophet's covenants with the Christians of Najran and with the Jews of Khaybā and Maqnā explicitly prohibited the expulsion of these communities from the Arabian Peninsula.[168] According to Mālik, al-Zuhrī had heard that the Prophet said "No two religions shall co-exist in the Arabian Peninsula."[169] 'Abd al-Razzāq reports a similar tradition in which he mentions how al-Zuhrī obtained his report from Sa'īd b. al-Musayyib. 'Abd al-Razzāq's report also appears to be a compromise to that of Mālik when he expresses uncertainty as to whether it was in the land of the Arabs or in the Ḥijāz that two religions shall not co-exist.[170] A telling report in the Muwaṭṭa' has 'Umar b. 'Abd al-'Azīz say: "The last words which the Messenger of Allah—peace and blessings be upon him—uttered were: 'May Allah destroy the Jews and the Christians! They took the graves of their prophets as places of worship. No two religions shall remain in the land of the Arabs.'"[171] This report seems to be a strong indicator that the policy of expelling the Jews and Christians from the Arabian Peninsula was instigated by 'Umar b. 'Abd al-'Azīz and that this was in fact his statement, not that of the Prophet.

There may have been a deliberate attempt to project the policies of 'Umar b. 'Abd al-'Azīz to his greater namesake, 'Umar b. al-Khaṭṭāb. The voluntary relocation of many Jews and Christians in the time of 'Umar I would have enabled traditionists in the court of 'Umar II to re-envisage them as expulsions. A tradition going back to the Common Link Ibrāhīm b. Maymūn with the family isnād Sa'd b. Samura–Samura b. Jundub–Abū 'Ubayda b. al-Jarrāḥ informs us that, "The last words that the Prophet—peace and blessings be upon him—uttered were: 'Expel the Jews of the Ḥijāz from the Arabian Peninsula.'"[172] A variant of this same tradition has the Prophet add: "and know that the worst of people are those who take their graves as places of worship'"[173], while another variant has him instruct his followers to also expel the people of Najran.[174] It is noteworthy how Abū Yūsuf's recension of Abū Bakr's *Compact with the Christians of Najran*[175] makes no mention of the people of Najran's right to reside in the Arabian Peninsula, reflecting an earlier attitude when their presence in Arabia was not an issue. Al-Ṭabarī's version on the other hand records how their protection is enduring except for what was revoked by the Prophet, based on the command of Allah, "over their land and the land of the Arabs that no two religions shall dwell therein."[176]

Abū al-Zubayr al-Makkī is the Common Link to a number of traditions commanding the expulsion of the Jews and Christians from the Arabian Peninsula. He tells us that Jābir b. 'Abdullāh informed him that the Prophet said, "If I am to live long enough, I shall expel the Jews and the Christians from the Arabian Peninsula until I only have Muslims remaining there."[177] A tradition on the authority of Ibn 'Abbās in which he recounts the Calamity on Thursday has him attest that the Prophet made a bequest concerning three matters, the first of which was to expel the polytheists (not the People of Book) from the Arabian Peninsula; the second was for his successors to honor delegations in the same way that he did; and the third was either not mentioned by Ibn 'Abbās or forgotten by the narrator

168 (El-Wakil 2016), "The Prophet's Treaty with the Christians of Najran," pp. 320–5.
169 (Mālik 1985), *Muwaṭṭāa'*, vol. 2, Book 45, ḥadīth No. 18, p. 892.
170 (al-Ṣan'ānī 1983), *Muṣannaf*, vol. 10 ḥadīth No. 19359, p. 357.
171 (Mālik 1985), *Muwaṭṭā'*, vol. 2, Book 45, ḥadīth No. 17, p. 892.
172 (al-Ṭayālsī 1999). Also see (Abū 'Abdullāh Muḥammad b. Yasīl al-Ḥumaydī 1992).
173 (Ibn Ḥanbal 2001), *Musnad*, vol. 3, ḥadīth No.1691, p. 221; ḥadīth No. 1694, p. 223.
174 Ibn Ḥanbal, *ibid.*, ḥadīth No. 1699, p. 227.
175 (Abū Yūsuf 1999), *Kitāb al-Kharāj*, p. 73.
176 (Muḥammad b. Jarīr al-Ṭabarī 1879), *Tārīkh*, vol. 2, p. 535.
177 (al-Ṣan'ānī 1983), *Muṣannaf*, vol. 10, ḥadīth No. 19365, p. 359; and vol. 6, ḥadīth No. 9985, p. 54. Also see (Ibn Ḥanbal 2001), *Musnad*, vol. 1, ḥadīth No. 201, p. 329; ḥadīth No. 219, p. 343; (Muslim 2006), *Ṣaḥīḥ*, vol. 2, ḥadīth No. 63, p. 846; (al-Tirmidhī 1996), *Jāmi'*, vol. 3, ḥadīth No. 1607, p. 253.

Sa'īd b. Jubayr.[178] All of these traditions seem to be reflective of late Umayyad policy and having very little to do with the Prophet or the early Caliphs of Islam.

*7.6. Two Creeds Cannot Inherit One Another*

The Musnad of Zayd reports how 'Alī had heard the Prophet say, "People of two different creeds cannot inherit one another."[179] This ḥadīth is authentic but the original context does not seem to have anything to do with the rights of inheritance. The origins of this tradition can be found in 'Alī's *Covenant with the Magi* when he writes: "I have preserved the tradition of those born into the Magian religion to follow the Magian leadership when I heard the Messenger of Allah, peace and blessings be upon him, say: 'People of two different creeds shall not inherit one another.'"[180] The ḥadīth was also allegedly narrated by Ja'far al-Ṣādiq in *al-Kāfī* and in *Da'ā'im al-Islām* in the context of inheritance using wording that is very close to what we find in *'Alī's Covenant with the Magi*. The Sunni compilations also report similar ḥadīths[181], though the original intent of the tradition appears to have been the rejection of religious syncretism, not inheritance.

## 8. Compromising the Terms and Conditions of the Original Covenants

One would wish to think that the sole reason early Islamic scholars did not record the covenants in their books is because they were ignorant of them or out of caution not to transmit erroneous documents that were in the hands of non-Muslims. Though this would certainly have been the case on the part of sincere scholars, the sad truth of the matter suggests that there was a deliberate attempt to reverse the terms and conditions of the original covenants through the use of fabricated ḥadīths and fictitious isnāds. Though it would seem the transmission of the compacts relied on knowledge of the covenants, their tonality indicates that they are corrupted versions of the originals, reflecting instead the sentiments of late Umayyad and 'Abbasīd Caliphs in the way they conducted their affairs with their non-Muslim subjects.

Abū Yūsuf's recollection of the contents of the original treaties that were issued during the Islamic conquests reflects a compromising attitude that is in conformity to the state policy. He explains to the Caliph Hārūn al-Rashīd in his *Kitāb al-Kharāj* that the early Muslims had drafted treaties with the local populations to protect their synagogues and churches, their livelihoods, and guaranteeing that they would fight their enemies. He notes how "virtually the whole of Greater Syria and al-Ḥīra were conquered because of this and this is why their synagogues and churches were not destroyed."[182] Abū Yūsuf acknowledges that a number of treaties were formulated by Khālid b. al-Wālid[183] and Abū 'Ubayda b. al-Jarrāḥ[184] but the versions he recounts differ in tonality from the authentic covenants of the Prophet and of 'Umar. According to Eutychius, all covenants issued to the people of Greater Syria, Palestine, and Jordan were modeled on the *Treaty with the People of Damascus*, which did not contain any of the discriminatory clauses Abū Yūsuf describes, meaning that Khālid and Abū 'Ubayda were following the tolerant policy of 'Umar.[185] Abū Yūsuf was clearly trying to compromise among competing Muslim attitudes of the other as he himself elaborates:

---

178  (al-Ṣan'anī 1989). Also see (al-Ṣan'anī 1983) *Muṣannaf*, vol. 6, ḥadīth No. 9992, p. 57; (al-Bukharī 2002), *Ṣaḥīḥ*, Book 64, ḥadīth No. 4431, p. 1087; Book 56, ḥadīth No. 3035, p. 702; and Book 58, ḥadīth No. 3168, p. 782; (Muslim 2006), *Ṣaḥīḥ*, Book 25, ḥadīth No. 20, p. 772; and (Abū Dāwūd 2009), *Sunan*, vol. 4, Book 14, ḥadīth No. 3029, p. 640.
179  Zayd b. 'Alī 1999, *Musnad*, 332.
180  (El-Wakil 2017), "Searching for the Covenants," p. 133, sct. 8, 9.
181  For a good discussion of this ḥadīth, see (al-Asqalānī 1995).
182  (Abū Yūsuf 1999), *Kitāb al-Kharāj*, p. 152.
183  Abū Yūsuf, *ibid.*, p. 160.
184  Abū Yūsuf, *ibid.*, p. 152.
185  To read the *Treaty with the People of Damascus* and how it was a model for all subsequent treaties see (Eutychius 1909). Similar versions of the *Treaty* have been documented by Ibn 'Asākir. See Ibn 'Asākir, *Tārīkh Madīnat Dimashq*, vol. 2, pp. 117–18; pp. 180–81; pp. 354–55; vols. 6, 59; p. 225.

> "I do not consider it appropriate to annul anything of what was agreed in the treaties nor any attempts to terminate what Abū Bakr, ʿUmar, ʿUthmān and ʿAlī—may Allah be pleased with them all—agreed with them. Rather their policy should be maintained for they did not rescind on anything which was agreed with them in the treaties that they concluded. As for any new synagogues and churches that are built, then these should be destroyed. This was the opinion of more than one Caliph and those who preceded us.

> Some Muslims had sought to destroy the synagogues and churches in the [newly conquered] cities and provinces but the people of those cities took out their treaties in which these agreements were made between them and the Muslims. The jurists and the Successors reprimanded those Muslims who had considered destroying their synagogues and churches until they no longer entertained the idea.

> The truce is applicable based on what ʿUmar–may Allah be pleased with him—stipulated with them, that it is applicable until the Day of Judgment." [186]

The fact that Abū Yūsuf mentions how the truce is applicable until the Day of Judgment demonstrates strong awareness of the original covenants of the Prophet and of ʿUmar either by him or his informants. Though Abū Yūsuf advocated a certain degree of tolerance towards non-Muslim subjects, he nevertheless believed that their rights should be curtailed to safeguard prevailing state policy. As Milka Levy-Rubin has argued, "Abū Yūsuf's main goal seems to have been to be a form of coexistence that would be satisfactory to both sides,"[187] with his position being a reaction to more intolerant views. Discriminatory legislation against non-Muslims eventually culminated in the forged *Pact of ʿUmar* which A.S. Tritton noted "was drawn up in the schools of law"[188] and which Mark Cohen brilliantly remarked "incorporates features that are characteristic, not of the conquest situation, but of administrative procedures of the developed Muslim state."[189]

Though the discriminatory measures mentioned by Abū Yūsuf and other jurists are part of a developed legal framework, it should be noted that the historical reality was far more complex. The restrictions imposed upon the dhimmīs were not always officially imposed, and they were often of a symbolic nature rather than of practical import. Countless decrees were issued by Muslim rulers attesting the authenticity of the Prophet's *Covenant with the Monks of Mount Sinai*, the earliest of which can be dated to 502 AH/1109 CE.[190] Ṣalāḥ al-Dīn al-Ayyūbī is also said to have issued a covenant to the Armenian Patriarchate of Jerusalem on 6 October AD 1186/20 Rajab 582 AH when he entered the Holy City, reaffirming the rights that the Prophet, ʿUmar, and ʿAlī had granted them.[191] An inscription clearly influenced by the covenants by Mamlūk Sultan al-Zāhir Sayf ad-Dīn Jaqmaq (died 1453 CE/857 AH) at the entrance of the Armenian monastery of St. James in Jerusalem, and which was recorded in the official registers of the year 854 AH/circa 1450 CE, reads "May God's curses fall upon and follow, till the end of time, whosoever imposes a tribute or inflicts an injustice."[192]

The Ottomans issued numerous fermāns to the religious minorities under their jurisdiction. In Shawwāl 862 AH/August–September 1458 CE, Sultan Mehmed II wrote a fermān to the Patriarch of Jerusalem stating:

> "It is my most imperial command that those who have command over the affairs in my realm, be it by land or by sea, shall protect and preserve the Patriarch of Jerusalem and

---

[186]	(Abū Yūsuf 1999), *Kitāb al-Kharāj*, pp. 160–161.
[187]	(Levy-Rubin 2011).
[188]	(Tritton 1930, p. 233).
[189]	(Cohen 1999, p. 129).
[190]	(Library of Congress n.d.). Also see (Stern 1964; Jiwa 2009).
[191]	(Dadoyan 2013). For a brief discussion see (Ghazarian 2008, pp. 66–68). Ghazarian produced images of the covenants of the Prophet, ʿAlī, and Ṣalāḥ al-Dīn.
[192]	(Ghazarian 2008), "Armenians in Islamicjerusalem," p. 69.

the aforementioned monks from molestation by anyone; should anyone, be he one of my successors, or one of my high ministers, or one of the ulema, or some civil authority, or one of the slaves of my court, or anybody else from the Muslim community, wish, for money or favour, to annul [this command], may he encounter the wrath of God and His revered Prophet!" [193]

The fermān was palpably influenced by the covenants of the Prophet of which Sultan Mehmed II would have clearly been aware. Sultan Selim I, who brought the *Covenant of the Prophet with the Monks of Mount Sinai* back to Istanbul in 1517 CE[194], also issued his own fermān to the Patriarch of Jerusalem in conformity with it.[195] It is noteworthy that, recently, the General Directorate of State Archives of the Prime Ministry of the Republic of Turkey had on public display a copy of the *Covenant of 'Umar with the Christians of Jerusalem* which was copied out into one of their church registers in 1171 AH/circa 1758 CE, showcasing Ottoman tolerance of the other.[196] A comprehensive book was published in 2002 CE by Father Gabriel Akyüz demonstrating how Ottoman Sultans time and again issued fermāns to protect the Syriac Orthodox Christians in accordance with the covenants of the Prophet and of 'Umar.[197] Though more work needs to be done on the link between the Prophet's covenants and the fermāns issued by the Ottoman sultans, there is sufficient evidence from the historical record to indicate that the covenants were not unequivocally disregarded but that they were on more than one occasion considered legitimate by Muslim rulers.

## 9. Incorporating the Covenants into Islamic Law

Islamic law is derived from the Qur'ān and the sunna of the Prophet, and though the sunna has traditionally been understood to be the ḥadīths and the inherited living practice ('amal) of the Muslim community, the covenants suggest that at its very core it represented the Prophet's official decrees. Had the Prophet left the Qur'ān and the sunna behind as a source of guidance for future posterity, then he could not have possibly referred to ḥadīths which were collected many years after his death. Islamic history seems to have consequently witnessed the gradual replacement of official legislature by the ḥadīths, with some traditions even suggesting that the Prophet had prohibited his followers from writing anything other than the Qur'ān.[198] However, even if we are to assume that this was the case, then presumably it was most likely a prohibition to avoid his official decrees being confused with other writings.

As attested by the Jerusalem 32 inscription and John Bar Penkaye, the covenants played a major role in guiding state policy during the early Caliphate. Meïr Bravmann's excellent study persuasively argued that the early Islamic state possessed state archives[199], which means that a record of the Prophet's official decrees must have existed at some point in time. We can only assume that the archives were destroyed early on, perhaps during the second civil war, for it is indeed greatly surprising how none of the legal schools ever seem to stress their importance. Even Mālik's seemingly solid legal methodology which uses the ḥadīths to support the inherited "living practice"—with the two being

---

[193] Quoting from (Çolak 2013).
[194] For a brief overview of the history of the Monastery of St. Catherine, including the Prophetic covenant, see (Atiya 1952).
[195] (Çolak 2013), "Relations between the Ottoman Central Administration and the Greek Orthodox Patriarchates," pp. 53–54.
[196] The General Directorate of State Archives of the Prime Ministry of the Republic of Turkey (now renamed Republic of Turkey Presidency of State Archives) no longer has the exhibition on display; however, the image and text of the document can be consulted at Lastprophet.info, (Treaty for Quds) The Jerusalem Covenant was extracted from the Ottoman Archives in Istanbul, *The Church Registers*, No. 8, p. 5.
[197] See (Akyüz 2002), Osmanlı Devletinde Süryani Kilisesi.
[198] For example, one famous tradition states: "Do not write anything from me, for whoever of you writes anything from me other than the Qur'ān let him erase it. Narrate traditions from me instead for there is no harm in that, but whoever lies about me—the narrator Hammām added the word 'intentionally'—then he will find his place in the fire of hell." See (Muslim 2006), *Ṣaḥīḥ*, Book 53, ḥadīth No. 72, p. 1366.
[199] (Bravmann 2012).

a "strong confirmation"[200] of the original sunna—makes no explicit reference to official decrees or remains of state archives. It is very curious that despite Mālik's keenness to preserve the sunna of the Prophet in Madīna he neither records the *Constitution of Madīna* in its unabridged form nor a single official decree of the Prophet and the Rightly Guided Caliphs in his Muwaṭṭaʾ. There may of course be reasons for this, though at this point in time there are no clear answers to explain how ḥadīths difficult to reconcile with the covenants came about in his Muwaṭṭaʾ.

Instead of viewing the Qurʾān as prima scriptura by trying to understand its meanings holistically, with parts of it interpreting other parts, jurists argued that certain verses of the Qurʾān were abrogated, even going so far as to claim that the ḥadīths can abrogate the Qurʾān.[201] Additionally, they seem to have paid little attention to trying to understand the Qurʾān as a response to the major religious and intellectual debates taking place in its 7th century cultural milieu. Though it never actualized, the precedent of classifying the Qurʾān as prima scriptura is supported by a number of reports attributed to the Prophet, such as: "There will be people after me who will narrate ḥadīths, therefore assess what they narrate in relation to the Qurʾān taking from them whatever agrees with it and rejecting whatever does not agree with it." [202] Another tradition states: "O people! Whatever reaches you from me that agrees with the Book of Allah, I have said. Whatever disagrees with it, I have not said."[203]

The methodology of rejecting traditions which contradict the Qurʾān was outlined by the Imāms of ahl al-bayt. The famous jurist Jaʿfar al-Ṣādiq is reported to have said: "Whatever tradition reaches you, regardless of whether it be from a righteous man or from a reprobate, and it agrees with the Book of Allah, take it. Whatever contradicts the Book of Allah, be it be from a righteous man or from a reprobate, reject it."[204] He also said: "Whatever ḥadīth reaches you on our authority and which cannot be validated by the Book of Allah is void."[205] The muʿtazila, also advocated a rationalist approach to assessing the veracity of the ḥadīths. Abū al-Qāsim al-Kaʿbī (died 931 CE/319 AH) explained how due diligence should be applied to what the traditionists narrate except "for what does not contradict the Book of Allah in which there is no falsehood from beginning to end, and the sunna of the Messenger of Allah which has been agreed upon."[206]

The rejection of problematic ḥadīths due to textual incongruities has a precedent in Islamic history. This is clearly evidenced by how certain schools of law discarded ṣaḥīḥ traditions with multiple chains of transmission in favor of a popular practice or an isolated tradition which they deemed more authoritative.[207] The fact that scholars from different schools of law set their own distinct criteria for incorporating a ḥadīth within the fabric of Islamic law showcases that there never was a consensus on the isnād system being an absolute in validating a tradition's authenticity.

When we compare the ḥadīths to the Prophet's official decrees (i.e., the *Constitution of Madīna*, the covenants, and authentic compacts such as the Prophet's *Compact with al-ʿAlāʾ b. al-Ḥaḍramī*), we find that some agree with these official decrees while others contradict them. As the official decrees reflect the official policy of the early Islamic state, precedence ought to, in theory, be given to them if

---

[200] (Dutton 1993).

[201] For an excellent discussion of the theory of abrogation, see (Fatoohi 2013).

[202] (al-Dāraqutnī 2001). Al-Dāraqutnī mentions an alternative isnād for this tradition that reads ʿĀsim b. Abī al-Nujūd > Zayd b. ʿAlī > ʿAlī b. al-Ḥusayn. The ḥadīth is not raised to the Prophet, though its isnād is one of the most reliable. There is reason to deny this tradition, for the Prophet's Household always advised their followers to weigh a ḥadīth's veracity in relation to the Qurʾān.

[203] (al-ʿAyāshī 1991). Al-ʿAyāshī reports similar traditions on the authority of ʿAlī, al-Ḥusayn, Muḥammad al-Bāqir, and Jaʿfar al-Ṣādiq, see pp. 19–20.

[204] al-ʿAyāshī, *ibid*., p. 20.

[205] al-ʿAyāshī, *ibid*.

[206] (al-Kaʿbī 2000).

[207] Examples of this are too numerous to cite, though we may here draw the reader's attention to the mutawātir tradition permitting a Muslim to wipe his leather socks during ablution and which has been reported by 66 authorities, see (al-Kattānī n.d.). Despite its mutawātir status, this practice was rejected by the Ḥanafīs, the Zaydīs, the Jaʿfarīs, and the Ismāʿīlīs. The same applies for the case of sadl al-yadayn, which was adopted by the Mālikis based on the practice of the people of Madīna, and by the Zaydīs, the Jaʿfarīs, and the Ismāʿīlīs because of traditions they traced back to the ahl al-bayt which they considered more authoritative than the mutawātir traditions going back to the Prophet supporting that the hands be held together in prayer. For a good discussion see (Dutton 1996).

we postulate that the sunna of the Prophet was primarily understood to be his official legislature. The prominence of the official decrees as part of the sunna of the Prophet could, therefore, be supported by the following tradition: "There will reach you many ḥadīths about me, so whatever agrees with the Book of Allah and my sunna [i.e., official decrees] is from me and whatever contradicts the Book of Allah Most-High and my sunna [i.e., official decrees] is not from me."[208]

The veracity of certain ḥadīths can therefore be determined by their harmony to texts which are contemporary witnesses of the early Islamic state, namely, the Qurʾān and the Prophet's official decrees. If we refer back to the ḥadīth criticized by Goldziher in which the Prophet warned that anyone who lies intentionally about him will find his place in hell,[209] we know from a comprehensive reading of the Qurʾān that lying about religious matters is a grave sin, and from the covenants that the Prophet issued staunch warnings to those who would alter them. The authenticity of the tradition can, thus, be deduced from its matn, not its isnād, even though the latter should not be ignored as it could shed light on how the variants of the ḥadīth were transmitted. As for textually flawed compacts, we can, to a certain extent, obtain a gist of the tone of the original documents by cross-comparing their language to that of the covenants.

## 10. Conclusions

The rediscovery of primary documents dating from the Prophet and the Rightly Guided Caliphs' time poses difficult questions as to how Islamic history came to be written down. The authentic tradition "Whoever harms a dhimmī I shall be his foe on the Day of Judgment" does not have multiple chains of transmission to back it up, has not been recorded in any of the ṣaḥīḥ works, and has not been transmitted through any of the most highly esteemed isnāds. Our analysis of this tradition has also demonstrated how the theory of the Common Link as the most likely originator of a tradition cannot be entirely substantiated and that fabricated texts were oftentimes reproduced using complete isnāds to back a particular doctrine. Lastly, our analysis has revealed that authentic teachings of the Prophet are completely missing from the ḥadīth corpus while sayings that contradict his instructions actually found their way into the ḥadīth literature and in legal works.

Though the ḥadīth literature embodies authentic sayings of the Prophet, the isnād should not be viewed as the main criterion for assessing a tradition's genuineness. Analysis of the matn rather than reliance on the isnād should be the driving factor in testing a tradition's veracity so that it can be weighed against the Qurʾān and the Prophet's official decrees—of which the covenants are of central importance—to determine whether or not it should be incorporated within the fabric of Islamic law.

As it stands, scholars can no longer limit themselves to taking the canonical materials in Muslim sources at face value for a comprehensive reconstruction of early Islamic history. An inter-textual analysis of Islamic sources along with other materials from the same time period is very much needed. As this paper has attempted to demonstrate, an inter-textual reading of multiple sources offers insights into how the Islamic tradition has been constructed, redacted, and revised over time. A richer understanding of reality therefore requires such an inter-textual approach to hopefully offer a more promising path for the future of interfaith relations.

**Acknowledgments:** I would like to thank Ibrahim Zein from the College of Islamic Studies, Hamad Bin Khalifa University, for all of his support. I would also like to thank John Andrew Morrow for having initiated the study of the covenants of the Prophet and for the many invaluable references which he brought to my attention when I first came to write this paper. Finally, I would like to thank Hanane Bensaid for assisting me with the references. All errors remain mine.

**Conflicts of Interest:** The author declares no conflict of interest.

---

[208] (al-Baghdādī 1357 AH).

[209] Hansu explains how, by the time of al-Kattānī, the definition of mutawātir itself changed and that, previously, this tradition was classified by al-Tirmidhī as ṣaḥīḥ ḥasan gharīb. See (Hansu 2009), "Notes on the Term Mutawātir and its Reception in Ḥadīth Criticism," p. 404.

## Appendix A. Cross-Comparison of the Witnesses' Names in the Christian Covenants

| Gabriel[210]/Aubert[211] Covenant | Graf[212]/Akyüz[213] Covenant | Hill[214]/Morrow[215] Covenant |
|---|---|---|
| MATCHING NAMES | | |
| 1. Abū Bakr al-Ṣiddīq | 1. Ibn Bakr/Abū Bakr | |
| 2. ʿUmar b. al-Khaṭṭāb al-Farūq | 2. ʿUmar b. al-Khaṭṭāb | 1. Abū Bakr al-Ṣiddīq |
| 3. ʿUthmān b. ʿAffān | 3. ʿUmar b. ʿAffān/ʿUthman b. ʿAffān | 2. ʿUmar b. al-Khaṭṭāb |
| 4. ʿAlī b. Abī Ṭālib | 4. ʿAlī Abā Ṭālib/ʿAlī Abū Ṭālib | 3. ʿUthmān b. ʿAffān |
| 5. Abū Ḥurayra | 5. Abū Ḥurayra | 4. ʿAlī b. Abī Ṭālib |
| 6. ʿAbdullāh b. Masʿūd | 6. ʿAbdullāh b. Masʿūd | 5. Abū Ḥurayra |
| 7. Saʿd b. Muʿād | 7. Saʿd b. Muʿādh | 6. ʿAbdullāh b. Masʿūd |
| 8. Saʿd b. ʿUbada/Saʿīd b. ʿUbada | 8. Saʿd b. ʿĀn/Saʿd b. ʿAsada (i.e., ʿUbada) | 7. Saʿīd b. Muʿādh |
| 9. Hassān b. Thābit | | 8. Saʿīd b. ʿUbādah |
| 10. Abū Ward/Abū Zar | 9. Ḥassān b. Thābit | 9. Ḥassān b. Thābit |
| 11. Mulḥid b. ʿAbdullāh/Ṭalḥa b. ʿAbdullāh | 10. Abā Qārr/Abū Dharr | 10. Abū Darh/Abū Darr |
| | 11. Ṭalḥa b. ʿAbdullāh | 11. Ṭalḥah b. ʿAbdullāh |
| MATCHING NAMES WHICH ARE MOST LIKELY THE RESULT OF INCORRECT TRANSCRIPTION | | |
| 12. ʿAbbās al-Zuhīra/ʿAbbās al-Zuhawī | | 12. Faḍl b. al-ʿAbbās al-Zuhrī |
| 13. Yāsīn b. Qays | 12. Faḍl b. al-ʿAbbās/Fāḍil b. al-ʿAbbās | 13. Thābit b. Qays |
| 14. Umāmah b. Mulḥid/Usāma b. Badīr | 13. Tābit b. Ghayth | 14. Imāmr b. Yazīd/Umāmah b. Yazīd |
| 15. Sahl b. ʿUmar/Shahl b. ʿUmar | 14. Umāmī b. Zayd/ Umāmah b. Zayd | 15. Shahl b. Tamīm/Shahr b. Abī al-Murr |
| 16. ʿAbd al-ʿAẓīm b. Ḥusayn/ʿAbd al-ʿAẓīm b. Ḥasan | 15. Sahl b. Zād/Shahl b. Murād | 16. ʿAbd al-ʿAẓīm b. al-Ḥusayn/ʿAbd al-Muʿẓam b. al-Ḥusayn |
| | 16. ʿAbd al-ʿAẓīm b. Hayt/ʿAbd al-ʿAẓīm b. Hanīm | |
| MATCHING NAMES WITH MORE THAN ONE COVENANT | | |
| | | 17. Yazīd b. Talīt/Zayd b. Maktab |
| | | 18. ʿAbdullāh b. Yazīd/Abdullah Abī Yazīd |
| 17. Abū al-Wardān/Abū al-Wardā | | 19. Abū al-Dardā |
| 18. ʿAbdullāh b. ʿAbd al-Wāḥid | 17. Abī al-Ward/Abū al-Wadūd | 20. Farṣūṣ b. Qāsīm/Marṣūṣ b. Basm |
| 19. Abū Ṣūṣ b. Qāsim | 18. ʿAbdullāh b. ʿAbd al-Wāḥid | 21. Mīr b. Ibrahīm/Zayd b. Abrahām (Mistakenly conflated with Farṣūṣ b. Qāsīm in the Morrow recension) |
| 20. Mīr b. Ibrahīm | 19. Zayd b. al-Nakīb/Zayd b. Maktīb | |
| 21. Mulḥim b. Mūsā/Maʿẓam b. Mūsā | 20. ʿAbdullāh b. Zayd | |
| 22. Abū Ḥanīfa/Abū Ḥīfa | 21. Abū al-Dardā | 22. Maʿẓam b. Mūsā |
| 23. ʿId b. Manṣūr | | 23. Abū Ḥanīfa |
| 24. Hāshim b. ʿAbdullāh | | 24. ʿUbayd b. Manṣūr |
| | | 25. Hāshim b. ʿAbdullāh |

| Gabriel[210]/Aubert[211] Covenant | Graf[212]/Akyüz[213] Covenant | Hill[214]/Morrow[215] Covenant |
|---|---|---|
| | NAMES THAT DO NOT MATCH | |
| | 22. al-ʿAbbās | |
| | 23. ʿAwḍ b. Qāsim/ʿUways b. Qāsim | |
| 25. ʿAbdullāh b. ʿAbbās | 24. Zayd b. Arqam(?)/Zayd b. Ibrāhīm | 26. al-ʿAbbās b. ʿAbd al-Mālik |
| 26. Faḍl b. ʿUmar | 25. ʿUthmān b. Abā Ghafān/ʿUthmān b. Abā Ghamār | 27. ʿAbdullāh b. ʿAmr b. al-ʿĀṣ/ʿAbdullāh b. ʿAbd al-Mūʾmin |
| 27. Zayd b. Qalb | 26. Muṭʿim b. Aba Yūnus/Muʿẓam b. Mīrāy | 28. ʿAmar b. Yāsīr/ʿUthmān b. b. Yāsīr |
| 28. ʿAbdullāh b. Badīr/ʿAbdullāh b. Maryam | 27. Ibn Ṣafiyya/Ibn Ḥanīfa | 29. ʿAbū al-ʿĀzir |
| 29. ʿUthmān b. ʿId/ʿIzz | 28. ʿUqayl b. Muqrīn/ʿAqil b. Manṣūr | 30. Hishām b. ʿAbd al-Muṭṭalib |
| 30. Abū al-Ṣādir/Abū al-Nādir | 29. Hānīʾ b. ʿAbdullāh | |
| | 30. Ramaḍān b. ʿAbdullāh al-Ṭālib/Ramaḍān b. ʿAbd al-Ṭālib | |

## Appendix B. The Isnād Bundles of the Najran Compact

(a) Isnād Bundle 1

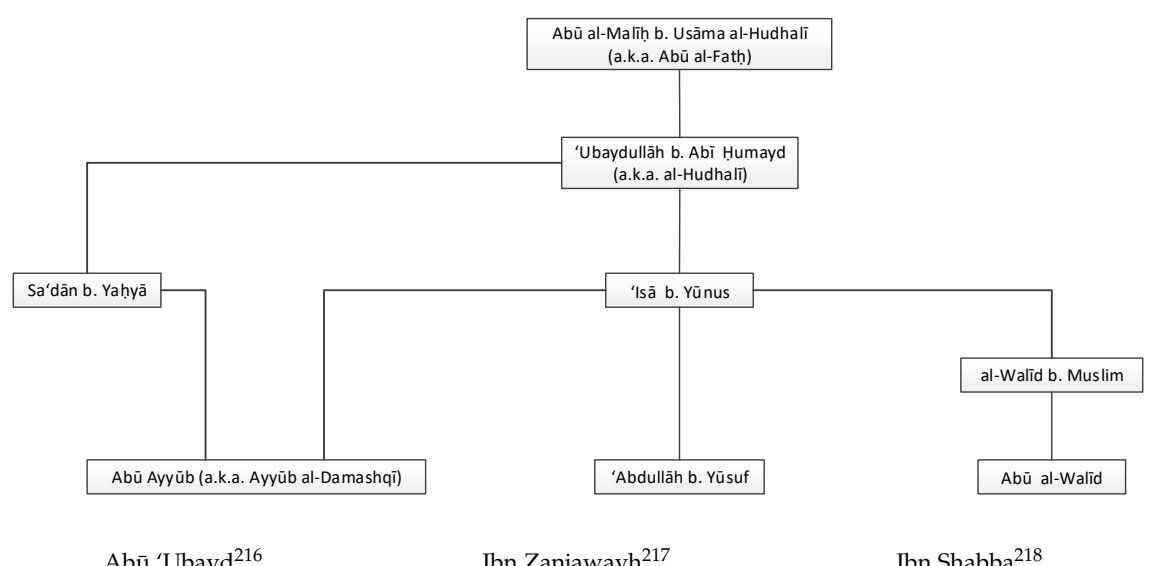

Abū ʿUbayd[216]          Ibn Zanjawayh[217]          Ibn Shabba[218]

210 (Aubert 1938), *Le Serment du Prophète*, pp. 40–41.
211 (Gabriel 1900), *Tārīkh al-Kanīsa*, pp. 593–94.
212 (Graf 1914), "Ein Schutzbrief Muḥammeds für die Christen," p. 566. Names have been deciphered from high-resolution images of the original manuscript, not Graf's transcriptions.
213 (Akyüz 2002), *Osmanlı Devletinde Süryani Kilisesi*, pp. 160–61.
214 (Ar. 202 2008), pp. 161b–162a).
215 (Morrow 2013), *Covenants*, pp. 262–63.
216 (Abū ʿUbayd 1989), *Kitāb al-Amwāl*, ḥadīth No. 504, pp. 280–81.
217 (Ibn Zanjawayh 1986), *Kitāb al-Amwāl*, ḥadīth No. 732, pp. 449–50.
218 (Ibn Shabba 1996).

(b) Isnād Bundle 2

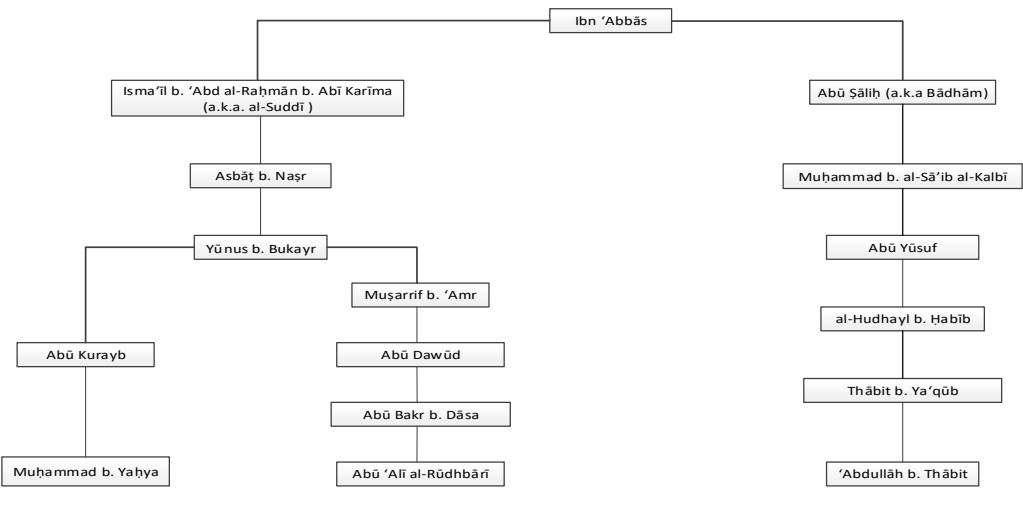

Abū al-Shaykh al-Iṣfahānī[219]　　　al-Bayhaqī[220]　　　Muqātil b. Sulaymān[221]

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
