# Peer review of "“Whoever Harms a Dhimmī I Shall Be His Foe on the Day of Judgment”: An Investigation into an Authentic Prophetic Tradition and Its Origins from the Covenants"

_religions, doi:10.3390/rel10090516_

Round 1

Reviewer 1 Report

This is an interesting piece that reads like a detective story. The article opens with a central question - why the covenants, which appear to have a strongly documented history, are not mentioned in the hadith and are not included as part of the Islamic canon or even as legal maxims, despite the provided evidence of their prominence during Muhammad's lifetime. The author(s) argue that, as with other known concerns with respect to the hadith literature, there is evidence of redaction with respect to this issue during the early Abbasid Empire, likely for political reasons. Detailed attention is given to the manuscript sources and evidence, providing corroboration of external evidence and reference to these covenants, rather than relying purely upon the inherited Islamic tradition. As such, it marks an important inquiry into application of the historical-critical approach to the covenants and the hadith.

Overall, the evidence and questions about that evidence - such as why there are slight variations yet a consistent overall framework, why Mu'awiya would have been cut out of the picture despite his proximity to Muhammad as one of his two most prolific scribes (the other having been Ali), and analysis and comparison of isnads and the role of certain Companions - is clearly presented and argued. The relevant literature in the field is thoughtfully and rigorously engaged, although I would note relatively heavy reliance upon the works of the Classical Orientalists (Schacht, Goldziher, Juynboll, etc.) and limited engagement with contemporary works on hadith (notably missing is Asma Afsaruddin), possibly due to the specific focus of the topic.

Concerns/Critiques:

Sometimes the evidence cited jumps around significantly in terms of time periods - use of a 13th or 16th century text as evidence of authenticity for a 7th century text is not historically convincing. A statement about such time gaps would bolster the academic argument - or at least explain why a particular later source is considered to be reliable as an authenticator. Along the same lines, I find it problematic to accept that a mention of a covenant (such as for the Magi) from a writer 3 or 4 centuries after Muhammad's lifetime constitutes solid evidence of its authenticity. The texts that are closest to Muhammad's lifetime seem to me to be the most convincing. It is not necessarily surprising that later texts might mention earlier texts, but without earlier mentions, it is difficult to accept them as authoritative. Given the insistence of the overall piece on the importance of written documents and the notations of scribes, it seems odd not to have documentation closer to the actual time period.

Missed opportunities to highlight significant points - example - the discussion about the 11 witnesses to the Covenants with the Christians compared to the 7 witnesses to the Covenant with the Magi. I noted that all 4 of the Rightly-Guided Caliphs were among the witnesses, which seemed to me to be VERY significant - yet it was not highlighted. If all 4 had routinely witnessed such documents as part of Muhammad's statecraft, it seems to me that following his example ought to have assured their continuation - and solidified this, particularly in the eyes of the Salafis who seek to follow the examples of Muhammad and the Successors. It becomes even more curious then that the Covenants do not enjoy more prominence - and certainly any deviation from them, such as is purported to have occurred under Umar, would become significantly more problematic. (Umar is known to have made other changes to Muhammad's own example, such as by banning women from praying in the mosque, which raises questions about his rule as a caliph in general, but that may digress too much from the article's intended purpose.)

This piece presents a strong challenge to the Muslim community with respect to the hadith and what is not contained within it that will likely prove to be sensitive. The authors convincingly argue that it is odd that none of these covenants are testified to in the hadith. Given the evidence presented of the existence of such covenants, it really does raise questions as to why the Muslim community has not pressed harder to recover these important parameters for interfaith interactions - and why this material, in particular, is not present in the hadith. This is an important question and it should be raised. I'm not certain, though, that one can really argue that the covenants essentially constitute an additional source of scripture that should hold equally authoritative status with the Qur'an and Sunna.

There was a statement that there is no evidence of abrogation of the covenants, which the author(s) take to assert as the absolute authority of the covenant model. I'm not sure that this claim is proven anywhere in the article - it is simply stated. One might counter-argue that there is a plethora of evidence of abrogation if the covenant model was not retained within the broad Islamic tradition and if individual rulers (including, apparently, the Rightly-Guided Caliphs) did not make use of the covenant model. The concept of abrogation in the Qur'an has strong documentation (and debate) in the legal and theological literature. I'm not certain that one can really apply the principle of abrogation to the covenantal model - covenants are not part of the Qur'anic model, but are part of the Sunna - perhaps. The driving question for a theologian might be whether Muhammad was acting merely as the head of state or as the Prophet of God in this case. Were covenants ALWAYS put in place (as a religious or legal maxim) or were they limited to situations of political praxis? It might be wise to provide some context for these covenants - and then investigate whether there were other situations where covenants were not granted. As it currently stands, the article gives the impression that there are ALWAYS supposed to be covenants with certain groups of people. The overall record sets parameters for such covenants - they are not inherent or to be taken for granted. Plus, covenants can also be broken if the terms are not abided by.

Author Response

Dear reviewer,

Thank you very much for having read this piece and for your comments. Your efforts are much appreciated. Please find below the response to the points you raised:

Point 1: Sometimes the evidence cited jumps around significantly in terms of time periods - use of a 13th or 16th century text as evidence of authenticity for a 7th century text is not historically convincing. A statement about such time gaps would bolster the academic argument - or at least explain why a particular later source is considered to be reliable as an authenticator. Along the same lines, I find it problematic to accept that a mention of a covenant (such as for the Magi) from a writer 3 or 4 centuries after Muhammad's lifetime constitutes solid evidence of its authenticity. The texts that are closest to Muhammad's lifetime seem to me to be the most convincing. It is not necessarily surprising that later texts might mention earlier texts, but without earlier mentions, it is difficult to accept them as authoritative. Given the insistence of the overall piece on the importance of written documents and the notations of scribes, it seems odd not to have documentation closer to the actual time period. 

I have tried to address this point by mentioning the descriptions we find in the sources concerning these documents. We are told that the Najran Covenant, the Mount Sinai Covenant, the Covenant with the Children of Israel, and the Covenant with the Magi all had the Prophet's seal on them. I have mentioned only these documents out of relevancy. For example the Siffin Arbitration Agreement is said to have had 'Ali and Mu'awiya's seals on it. The Prophet's letters to the various kings are also said to have had his seal on them. Without digressing, it is certainly of note that different religious communities report his official seal having existed on these covenants. I would therefore argue that these documents remained in their possession as sacred relics, and when books of tradition compiled or Muslim rulers decided to collect them, they recorded them in their works. We therefore find that the first time the Covenant with the Monks of Mount Sinai is recorded is after Selim I's expedition to Egypt. Numerous copies of this document then followed and were issued to Christian communities in the Ottoman empire. As for the Covenant with the Magi, we are told the original document remained with the descendants of Salman so there is no reason why we should not regard it as a relic of theirs in the same way as the Covenant with the Monks of Mount Sinai was recognized as a precious relic by the Ottomans.

Point 2: Missed opportunities to highlight significant points - example - the discussion about the 11 witnesses to the Covenants with the Christians compared to the 7 witnesses to the Covenant with the Magi. I noted that all 4 of the Rightly-Guided Caliphs were among the witnesses, which seemed to me to be VERY significant - yet it was not highlighted. If all 4 had routinely witnessed such documents as part of Muhammad's statecraft, it seems to me that following his example ought to have assured their continuation - and solidified this, particularly in the eyes of the Salafis who seek to follow the examples of Muhammad and the Successors. It becomes even more curious then that the Covenants do not enjoy more prominence - and certainly any deviation from them, such as is purported to have occurred under Umar, would become significantly more problematic. (Umar is known to have made other changes to Muhammad's own example, such as by banning women from praying in the mosque, which raises questions about his rule as a caliph in general, but that may digress too much from the article's intended purpose.) 

There are actually 14 witnesses to the Covenant with the Magi though 7 of them are repeated in the Christian covenants. I have included an inscription - Jerusalem 32 - as evidence that the protection of God and His messenger was granted to the conquered populations inspired by the covenants. Its date suggests that it reflects the sentiments of the Caliphs up until 'Utham. Furthermore, John Bar Penkaye is a first-hand eye witness that the Caliphs honoured Christianity up until the death of Yazid. I have mentioned the various covenants of 'Umar in Christian sources and they all read like the covenants of the Prophet. The covenant of Khalid b. al Walid to the people of Damascus and the covenant of 'Amr b. al-'As to the people of Egypt which were issued under the rule of 'Umar all guarantee the protection of their churches. I didn't go into the details about all the texts issued by 'Umar and his generals as this would take too long (though they are referenced in the footnotes). The point is that the covenants of 'Umar were corrupted and the most obvious example of this is the forged Pact of 'Umar which reads nothing like his original covenants. 

Point 3: I'm not certain, though, that one can really argue that the covenants essentially constitute an additional source of scripture that should hold equally authoritative status with the Qur'an and Sunna.

I have tried to answer this point. Essentially we know that written decrees were issued. There is obviously a big difference when a ruler says something to when he writes a decree, has it witnessed, dated and stamped. I do not argue that these Prophetic decrees are an additional source to the sunna but rather that they are part of the sunna. We could here argue we have three sources of sunna: the hadith, 'amal and the official decrees.   

Point 4: There was a statement that there is no evidence of abrogation of the covenants, which the author(s) take to assert as the absolute authority of the covenant model. I'm not sure that this claim is proven anywhere in the article - it is simply stated. 

I have tried to address this point. All of the covenants, including those of 'Umar state that they apply until the Day of Judgment. Also if the Prophet had say, abrogated his covenants before his death, why would 'Umar and 'Ali issue covenants that read exactly the same?

Point 5: One might counter-argue that there is a plethora of evidence of abrogation if the covenant model was not retained within the broad Islamic tradition and if individual rulers (including, apparently, the Rightly-Guided Caliphs) did not make use of the covenant model. The concept of abrogation in the Qur'an has strong documentation (and debate) in the legal and theological literature. I'm not certain that one can really apply the principle of abrogation to the covenantal model - covenants are not part of the Qur'anic model, but are part of the Sunna - perhaps. 

There is no evidence of this except for the forged Pact of 'Umar. In fact the Covenant of 'Umar with the Christians of Jerusalem states: "It was granted to them from the generous and beloved prophet sent by Allah, who honoured them with his [covenant] using his blessed hand when he ordered that their needs be catered for and that they be protected." The religious tolerance described by John Bar Penkaye also suggests they were not annulled by the Rightly-Guided Caliphs. As argued in the paper, the violation of these original treaties occurred during the late Umayyad period/early Abbasid period. The first Caliph who is known to have confiscated churches and disrespectful of Christians is al-Walid. The first jizya to be imposed on monks, according to Ibn al-Makin, was under the rule of 'Abd al-Malik.

Point 6:The driving question for a theologian might be whether Muhammad was acting merely as the head of state or as the Prophet of God in this case. 

I think this point opens the door to too much speculation which I would like to avoid. Some of them have the Prophet state that a particular injunction was revealed to him e.g.  that paradise longs for Salman in the Covenant with the Magi; that the Jews are to return to their homes in the Covenant with the Jews of Khaybar and Maqna; and that the Jews are place their zunnar on thei r turbans in the Covenant with the Children of Israel. 

Point 7: Were covenants ALWAYS put in place (as a religious or legal maxim) or were they limited to situations of political praxis? It might be wise to provide some context for these covenants - and then investigate whether there were other situations where covenants were not granted. As it currently stands, the article gives the impression that there are ALWAYS supposed to be covenants with certain groups of people. The overall record sets parameters for such covenants - they are not inherent or to be taken for granted. Plus, covenants can also be broken if the terms are not abided by. 

My article tries to focus on the rights of non-Muslims and the relationship that ought to exist between Muslims and non-Muslims living side by side with one another. The breach of the covenants would deal with a separate issue, namely declaration of war and rules of warfare. At that point the issues one would need to look at are the fate of the Banu Qaynuqa, Qurayza and Nadir, and why they differed, perhaps because the punishment was relative to their treachery. One would also need to look at the apostasy wars which would also have been interpreted as treachery. 

I would argue yes, starting with the Constitution of Madina. The covenants with Christians mention how the Christians are not to betray the Muslims by siding with their enemy. It also does seem that in the aftermath of war the non-Muslims were granted a covenant. Nevertheless, what was their fate? Were some of their places of worship confiscated? Were they killed or enslaved? I think we would be here dealing with the laws of warfare in Islam which is beyond the scope of this paper. The paucity of the sources on whether this happened, when this happened, and the consequences of breaching the covenants by both sides - in the first instance with communities that peacefully capitulated and in the second instance with communities that were fought - would sidetrack the arguments presented in this article. We would then be dealing with the rules of warfare which I would like to avoid.

Reviewer 2 Report

This is very good article, displaying high-quality research. The topic is very important and the article surveys a broad range of material related to the topic. The section before the conclusion which suggests that the isnāds should be disregarded in the assesment of historical narratives is prescriptive and needs to be substantiated further, at least, it should demonstrate how we can depart from the group of material analysed here to make the leap in generalising about the hadith literature as a whole. The charts in the appendix are useful, tries to display the isnads, however the transmission analysis remains on a rudimentary level, occasionally includes mistakes. Why isnads should be disregarded needs also to be discussed in more depth in the introductory parts. Some repeating spelling mistakes such as Abdullah instead of Abdullāh, have been marked in the digital copy of the paper. Man ẓalama turns into man ẓalam after a certain point. Muʿāwiya is correct but it can be written as Muʿāwīya as well. Aṣfaḥānī - Iṣfaḥānī/Iṣbahānī? Muwaṭṭaʾ > Muwaṭṭāʿ etc.

Author Response

Dear reviewer,

Thank you very much for your useful comments and having taken the time to read the piece. Your efforts are much appreciated.

I have taken into account all of you points. Thank you very much for having pointed these out to me. In the revised version I try to emphasize giving priority to the matn rather than the isnad. I do not think it appropriate to argue that the isnads should be disregarded altogether.

Round 2

Reviewer 1 Report

Important revisions have been undertaken, including care not to overstate the case. References are clearly stated and source material seems comprehensive. This piece offers significant attention to primary sources and covers an impressive array of source materials, including those most relevant to the time period in question.

I have some concerns about accepting the veracity of the Sergius Bahira account as historical fact. Barbara Roggema's analytical work is cited, but her critical analysis doesn't seem to be incorporated.

I appreciated the attention to the content (matn) of hadith, rather than simply the chains of transmission (isnad). One part I found a bit confusing was on p. 18 where the author states the need to engage this kind of analysis and then proceeds to discuss the isnad. I recommend a clearer statement that both kinds of analysis can be undertaken, but isnad alone does not "prove" anything. Making that point clear in this section then points to the importance of the discussion that follows in which both methods are engaged and brought together.

Sometimes the level of detail is perhaps too much. Why is it important, for example, on p. 19 that a particular document was dated on a Monday? Cutting some of the superfluous detail would help the reader to remain focused on the argument. That said, I do appreciate that, in working to establish these covenants as central to the Islamic tradition that every bit of evidence available in support helps to bolster the argument. But I do think there are times when the level of detail is a bit too much.

Valuable contribution analyzing the redaction of the Umayyads, esp. Mu'awiyah, during the time of the Abbasids. Changes in political climate clearly had an impact on the development of what became "the Islamic tradition," so this is an important lens for thinking about texts, contexts and what has survived vs. what has not. The point about Mu'wayah and 'Ali having been among the Prophet's most important scribes is one well worth contemplating - it really does raise questions about why they do not appear more prominently in the foundational literature. Sometimes what is missing is as important as what appears.

One point the author might want to think about making in the introduction has to do with the methodology undertaken here, which is not stated as explicitly as it could be. Essentially, the author is undertaking an intertextual reading and analysis in this paper. Rather than simply looking at what have come to be canonical materials and accepting them at face value, the author is calling for an intertextual reading of these Islamic materials along with other materials from the same time period. It's not just about what Muslims have to say about Islam, but how other parties directly affected by Muslim expansion and rule had to say about Islam. That methodology is current in the field, although it remains in its early stages. Placing this article within that framework would bolster the arguments and place it within the field of scholarship within which the author is working more explicitly.

The conclusion falls flat. While I agree with the major points that have been documented about the lack of documentation within the Islamic canonical sources with respect to the covenants, the main thrust of the author's presentation of evidence is that one cannot simply look to those canonical sources for early Islamic history. The author has effectively shown that intertextual reading of multiple sources from this time period offers insights into how the tradition has been constructed, redacted, and revised over time, so that a richer understanding of reality requires such intertextual reading - and may offer a more hopeful path forward for future relations. That point should be made clearly at the conclusion.

This is a very important contribution and I look forward to seeing the final version published.

Author Response

Dear reviewer,

Thank you very much for your encouragement and very useful feedback. I have done my best to incorporate all of the points that you raised. Concerning the dating on a Monday, previous research on the covenants has argued that the day of the week in which they were written is one of the evidences for their authenticity. Calendar conversions show that indeed the date of the month and year in which they were written match the week day e.g. Monday or Friday in which which they were prupotedly issued. In the case of the Covenant with the Armenian Christians we know it is on a Monday in Dhu al-Hijja 2 AH, but the day of the month is unfortunately missing, meaning we don't know if it was the 1st, 2nd, 3rd... 29th. However for the sake of consistency I have added the detailed date to the letter to al-Alaa b. al-Hadrami.

Thank you once again for your insights and I wish you the very best. 
